# On the Convergence of
# Black-Box Variational Inference

**Kyurae Kim**
University of Pennsylvania
kyrkim@seas.upenn.edu

**Jisu Oh**
North Carolina State University
joh26@ncsu.edu

**Kaiwen Wu**
University of Pennsylvania
kaiwenwu@seas.upenn.edu

**Yi-An Ma**
University of California, San Diego
yianma@ucsd.edu

**Jacob R. Gardner**
University of Pennsylvania
jacobrg@seas.upenn.edu

## Abstract

We provide the first convergence guarantee for black-box variational inference (BBVI) with the reparameterization gradient. While preliminary investigations worked on simplified versions of BBVI (*e.g.*, bounded domain, bounded support, only optimizing for the scale, and such), our setup does not need any such algorithmic modifications. Our results hold for log-smooth posterior densities with and without strong log-concavity and the location-scale variational family. Notably, our analysis reveals that certain algorithm design choices commonly employed in practice, such as nonlinear parameterizations of the scale matrix, can result in suboptimal convergence rates. Fortunately, running BBVI with proximal stochastic gradient descent fixes these limitations and thus achieves the strongest known convergence guarantees. We evaluate this theoretical insight by comparing proximal SGD against other standard implementations of BBVI on large-scale Bayesian inference problems.

## 1 Introduction

Despite the practical success of black-box variational inference (BBVI; Kucukelbir *et al.*, 2017; Ranganath *et al.*, 2014; Titsias & Lázaro-Gredilla, 2014), also known as stochastic gradient variational Bayes and Monte Carlo variational inference, whether it converges under appropriate assumptions on the target problem have been an open problem for a decade. While our understanding of BBVI has been advancing (Bhatia *et al.*, 2022; Challis & Barber, 2013; Domke, 2019, 2020; Hoffman & Ma, 2020), a full convergence guarantee that extends to the practical implementations as used in probabilistic programming languages (PPL) such as Stan (Carpenter *et al.*, 2017), Turing (Ge *et al.*, 2018), Tensorflow Probability (Dillon *et al.*, 2017), Pyro (Bingham *et al.*, 2019), and PyMC (Patil *et al.*, 2010) has yet to be demonstrated.

Due to our lack of understanding, a consensus on how we should implement our BBVI algorithms has yet to be achieved. For example, when the variational family is chosen to be the location-scale family, the "scale" matrix can be parameterized linearly or nonlinearly, and both parameterizations are used by default in popular software packages. (See Table 1 in Kim *et al.* 2023.) Surprisingly, as we will show, seemingly innocuous design choices like these can substantially impact the convergence of BBVI. This is critical as BBVI has been shown to be less robust (*e.g.*, sensitive to initial points, stepsizes, and such) than competing inference methods such as Markov chain Monte Carlo (MCMC). (See Dhaka *et al.*, 2020; Domke, 2020; Welandawe *et al.*, 2022; Yao *et al.*, 2018.) Instead, the evaluation of BBVI algorithms has been relying on expensive empirical evaluations (Agrawal *et al.*, 2020; Dhaka *et al.*, 2021; Giordano *et al.*, 2018; Yao *et al.*, 2018).

37th Conference on Neural Information Processing Systems (NeurIPS 2023).

To rigorously analyze the design of BBVI algorithms, we establish the first convergence guarantee for the implementations *precisely* as used in practice. We provide results for BBVI with the reparameterization gradient (RP; Kingma & Welling, 2014; Titsias & Lázaro-Gredilla, 2014) and the location-scale variational family, arguably the most widely used combination in practice. Our results apply to log-smooth posteriors, which is a routine assumption for analyzing the convergence of stochastic optimization (Garrigos & Gower, 2023) and sampling algorithms (Dwivedi *et al.*, 2019, §2.3). The key is to show that evidence lower bound (ELBO; Jordan *et al.*, 1999) satisfies regularity conditions required by convergence proofs of stochastic gradient descent (SGD; Bottou, 1999; Nemirovski *et al.*, 2009; Robbins & Monro, 1951), the workhorse underlying BBVI.

Our analysis reveals that nonlinear scale matrix parameterizations used in practice are suboptimal: they provably break strong convexity and sometimes even convexity. Even if the posterior is strongly log-concave, the ELBO is not strongly convex anymore. This contrasts with linear parameterizations, which guarantee the ELBO to be strongly convex if the posterior is strongly log-concave (Domke, 2020). Under linear parameterizations, however, the ELBO is no longer smooth, making optimization challenging. Because of this, Domke (2020) proposed to use proximal SGD, which Agrawal & Domke (2021, Appendix A) report to have better performance than vanilla SGD with nonlinear parameterizations. Indeed, we show that BBVI with proximal SGD achieves the *fastest* known converges rates of SGD, unlike vanilla BBVI. Thus, we provide a concrete reason for employing proximal SGD. We evaluate this insight on large-scale Bayesian inference problems by implementing an Adam-like (Kingma & Ba, 2015) variant of proximal SGD proposed by Yun *et al.* (2021).

Concurrently to this work, convergence guarantees on BBVI with the RP and the sticking-the-landing estimator (STL; Roeder *et al.*, 2017) under the linear parameterization were published by Domke *et al.* (2023). To achieve this, they show that a quadratic bound on the gradient variance is sufficient to guarantee the convergence of projected and proximal SGD. In contrast, we focus on analyzing the ELBO under nonlinear parameterizations and connect it to existing analysis strategies. A more in-depth comparison of the two works is provided in Appendix E.

❶ **Convergence Guarantee for BBVI:** Theorem 3 establishes a convergence guarantee for BBVI with assumptions matching the implementations used in practice. That is, without algorithmic simplifications and unrealistic assumptions such as bounded domain or bounded support.

❷ **Optimality of Linear Parameterizations:** Theorem 2 shows that, for location-scale variational families, nonlinear scale parameterizations prevent the ELBO from being strongly-convex even when the target posterior is strongly log-concave.

❸ **Convergence Guarantee for Proximal BBVI:** Theorem 4 guarantees that, if proximal SGD is used, BBVI on $\mu$-strongly log-concave posteriors can obtain a solution $\epsilon$-close to the global optimum with $\mathcal{O}\left(1/\epsilon\right)$ iterations.

❹ **Evaluation of Proximal BBVI in Practice:** In Section 5, we evaluate the utility of proximal SGD on large-scale Bayesian inference problems.

## 2   Background

**Notation**   Random variables are denoted in serif (*e.g.*, $x$, $\mathbf{x}$), vectors are in bold (*e.g.*, $\boldsymbol{x}$, $\mathbf{x}$), and matrices are in bold capitals (*e.g.* $\boldsymbol{A}$). For a vector $\boldsymbol{x} \in \mathbb{R}^d$, we denote the inner product as $\boldsymbol{x}^\top \boldsymbol{x}$ and $\langle \boldsymbol{x}, \boldsymbol{x} \rangle$, the $\ell_2$-norm as $\|\boldsymbol{x}\|_2 = \sqrt{\boldsymbol{x}^\top \boldsymbol{x}}$. For a matrix $\boldsymbol{A}$, $\|\boldsymbol{A}\|_\mathrm{F} = \sqrt{\mathrm{tr}\left(\boldsymbol{A}^\top \boldsymbol{A}\right)}$ denotes the Frobenius norm. $\mathbb{S}_{++}^d$ is the set of positive definite matrices. For some function $f$, $\mathrm{D}_i f$ denotes the $i$th coordinate of $\nabla f$, and $\mathrm{C}^k\left(\mathcal{X}, \mathcal{Y}\right)$ is the set of $k$-time differentiable continuous functions mapping from $\mathcal{X}$ to $\mathcal{Y}$.

### 2.1   Black-Box Variational Inference

Variational inference (VI, Blei *et al.*, 2017; Jordan *et al.*, 1999; Zhang *et al.*, 2019) aims to minimize the exclusive (or backward/reverse) Kullback-Leibler (KL) divergence as:

$$\underset{\lambda \in \Lambda}{\text{minimize}} \; \mathrm{D}_{\mathrm{KL}}\left(q_\lambda, \pi\right) \triangleq \mathbb{E}_{\boldsymbol{z} \sim q_\lambda} - \log \pi\left(\boldsymbol{z}\right) - \mathbb{H}\left(q_\lambda\right),$$

where   $\mathrm{D}_{\mathrm{KL}}\left(q_\lambda, \pi\right)$   is the KL divergence,                    $\mathbb{H}$   is the differential entropy,
               $\pi$                 is the (target) posterior distribution, and   $q_\lambda$   is the variational distribution,

While alternative approaches to VI (Dieng *et al.*, 2017; Hernandez-Lobato *et al.*, 2016; Kim *et al.*, 2022; Naesseth *et al.*, 2020) exist, so far, exclusive KL minimization has been the most successful. We thus use "exclusive KL minimization" as a synonym for VI, following convention.

Equivalently, one minimizes the negative *evidence lower bound* (ELBO, Jordan *et al.*, 1999) $F$:

$$\underset{\lambda \in \Lambda}{\text{minimize}} \; F(\lambda) \triangleq \mathbb{E}_{\boldsymbol{z} \sim q_\lambda} - \log p(\boldsymbol{z}, \boldsymbol{x}) - \mathbb{H}(q_\lambda),$$

where $\log p(\boldsymbol{z}, \boldsymbol{x})$ is the *joint likelihood*, which is proportional to the posterior as $\pi(\boldsymbol{z}) \propto p(\boldsymbol{z}, \boldsymbol{x}) = p(\boldsymbol{x} \mid \boldsymbol{z}) p(\boldsymbol{z})$, where $p(\boldsymbol{x} \mid \boldsymbol{z})$ is the likelihood and $p(\boldsymbol{z})$ is the prior.

## 2.2 Variational Family

In this work, we focus on the following variational family. ($\overset{\text{d}}{=}$ is equivalence in distribution.)

**Definition 1** (**Reparameterized Family**). Let $\varphi$ be some $d$-variate distribution. Then, $q_\lambda$ that can be equivalently represented as

$$\boldsymbol{z} \sim q_\lambda \quad \Leftrightarrow \quad \boldsymbol{z} \overset{\text{d}}{=} \mathcal{T}_\lambda(\boldsymbol{u}); \quad \boldsymbol{u} \sim \varphi,$$

is said to be part of a reparameterized family generated by the base distribution $\varphi$ and the reparameterization function $\mathcal{T}_\lambda$.

**Definition 2** (**Location-Scale Reparameterization Function**). $\mathcal{T}_\lambda : \mathbb{R}^d \to \mathbb{R}^d$ defined as

$$\mathcal{T}_\lambda(\boldsymbol{u}) \triangleq \boldsymbol{C}\boldsymbol{u} + \boldsymbol{m}$$

with $\lambda$ containing the parameters for forming the location $\boldsymbol{m} \in \mathbb{R}^d$ and scale $\boldsymbol{C} = \boldsymbol{C}(\lambda) \in \mathbb{R}^{d \times d}$ is called the location-scale reparameterization function.

The location-scale family enables detailed theoretical analysis, as demonstrated by (Domke, 2019, 2020; Fujisawa & Sato, 2021; Kim *et al.*, 2023), and includes the most widely used variational families such as the Student-t, elliptical, and Gaussian families (Titsias & Lázaro-Gredilla, 2014).

**Handling Constrained Support** For common choices of the base distribution $\varphi$, the support of $q_\lambda$ is the whole $\mathbb{R}^d$. Therefore, special treatment is needed when the support of $\pi$ is constrained. Kucukelbir *et al.* (2017) proposed to handle this by applying diffeomorphic transformation denoted with $\psi$, often called *bjectors* (Dillon *et al.*, 2017; Fjelde *et al.*, 2020; Leger, 2023), to $q_\lambda$ such that

$$\boldsymbol{\zeta} \sim q_{\psi, \lambda} \quad \Leftrightarrow \quad \boldsymbol{\zeta} \overset{d}{=} \psi^{-1}(\boldsymbol{z}); \quad \boldsymbol{z} \sim q_\lambda,$$

such that the support of $q_{\psi, \lambda}$ matches that of $\pi$. For example, when the support of $\pi$ is $\mathbb{R}_+$, one can choose $\psi^{-1} = \exp$. This approach, known as automatic differentiation VI (ADVI), is now standard in most modern PPLs.

**Why focus on posteriors with unconstrained supports?** When bijectors are used, the entropy of $q_\lambda$, $\mathbb{H}(q_\lambda)$, needs to be adjusted by the Jacobian of $\psi$ (Kucukelbir *et al.*, 2017), $\boldsymbol{J}_{\phi^{-1}}$. However, applying the transformation to $\pi$ instead of $q_\lambda$ is mathematically equivalent and more convenient. In fact, bijectors can be automatically incorporated into our notation by implicitly setting

$$p(\boldsymbol{x} \mid \boldsymbol{z}) = \widetilde{p}\big(\boldsymbol{x} \mid \psi^{-1}(\boldsymbol{z})\big) \quad \text{and} \quad p(\boldsymbol{z}) = \widetilde{p}\big(\psi^{-1}(\boldsymbol{z})\big) \big| \boldsymbol{J}_{\psi^{-1}}(\boldsymbol{z}) \big|,$$

such that $\widetilde{\pi}(\boldsymbol{\zeta}) \propto \widetilde{p}(\boldsymbol{x} \mid \boldsymbol{\zeta}) \widetilde{p}(\boldsymbol{\zeta})$, where $\widetilde{\pi}$ is the constrained posterior that we are actually interested in. Therefore, our setup in Section 2.1, where the domain of $\boldsymbol{z}$ is taken to be the unconstrained $\mathbb{R}^d$, already encompasses constrained posteriors through ADVI.

Lastly, we impose light assumptions on the base distribution $\varphi$, which are already satisfied by most variational families used in practice. (*i.i.d.*: independently and identically distributed.)

**Assumption 1** (**Base Distribution**). $\varphi$ is a $d$-variate distribution such that $\boldsymbol{u} \sim \varphi$ and $\boldsymbol{u} = (u_1, \ldots, u_d)$ with *i.i.d.* components. Furthermore, $\varphi$ is **(i)** symmetric and standardized such that $\mathbb{E}u_i = 0$, $\mathbb{E}u_i^2 = 1$, $\mathbb{E}u_i^3 = 0$, and **(ii)** has finite kurtosis $\mathbb{E}u_i^4 = k_\varphi < \infty$.

The assumptions on the variational family we will use throughout this work are collectively summarized in the following assumption:

**Assumption 2.** The variational family is the location-scale family formed by Definitions 1 and 2 with the base distribution $\varphi$ satisfying Assumption 1.

## 2.3 Scale Parameterizations

For the "scale" matrix $\boldsymbol{C}(\lambda)$ in the location-scale family, any parameterization that results in a positive-definite covariance $\boldsymbol{C}\boldsymbol{C}^\top \in \mathbb{S}_{++}^d$ is valid. However, for the ELBO to ever be convex, the entropy $\mathbb{H}(q_\lambda)$ must be convex, which requires the mapping $\lambda \mapsto \boldsymbol{C}\boldsymbol{C}^\top$ to be convex. To ensure this, we restrict $\boldsymbol{C}$ to (lower) triangular matrices with strictly positive eigenvalues, essentially, Cholesky factors. This leaves two of the most common parameterizations:

| **Definition 3 (Mean-Field Family.).** | **Definition 4 (Full-Rank Cholesky Family).** |
|---|---|
| $$C = D_\phi(s)$$ | $$C = D_\phi(s) + L,$$ |
| where the $d$ elements of $s$ forms the diagonal and $\lambda \in \Lambda$ such that | where the $d$ elements of $s$ forms the diagonal, $L$ is $d$-by-$d$ strictly lower triangular, and $\lambda \in \Lambda$ such that |
| $$\Lambda = \{(m, s) \mid m \in \mathbb{R}^d, s \in \mathcal{S}\}.$$ | $$\Lambda = \{(m, s, L) \mid m \in \mathbb{R}^d, s \in \mathcal{S}, \text{vec}(L) \in \mathbb{R}^{(d+1)d/2}\}.$$ |

Here, $S$ is discussed in the next paragraph, $D_\phi(s) \in \mathbb{R}^{d \times d}$ is a diagonal matrix such that $D_\phi(s) \triangleq$ diag $(\phi(s)) = \text{diag}(\phi(s_1), \dots, \phi(s_d))$, and $\phi$ is a function we call a *diagonal conditioner*.

**Linear v.s. Nonlinear Parameterizations** When the diagonal conditioner is a linear function $\phi(x) = x$, we say that the covariance parameterization is *linear*. In this case, to ensure that $C$ is a Cholesky factor, the domain of $s$ is set as $\mathcal{S} = \mathbb{R}^d_+$. On the other hand, by choosing a nonlinear conditioner $\phi : \mathbb{R} \to \mathbb{R}_+$, we can make the domain of $s$ to be the unconstrained $\mathcal{S} = \mathbb{R}^d$. Because of this, nonlinear conditioners such as the softplus $(x) \triangleq \log(1 + \exp(x))$ (Dugas *et al.*, 2000) are frequently used in practice, especially for mean-field. (See Table 1 by Kim *et al.*, 2023).

### 2.4 Problem Structure of Black-Box Variational Inference

Exclusive KL minimization VI is fundamentally a composite (regularized) optimization problem

$$F(\lambda) = f(\lambda) + h(\lambda), \qquad \text{(ELBO)}$$

where $f(\lambda) \triangleq \mathbb{E}_{z \sim q_\lambda} \ell(z)$ is the *energy term*, $\ell(z) \triangleq -\log p(z, x)$ is the negative joint log-likelihood, and $h(\lambda) \triangleq -\mathbb{H}(q_{\psi, \lambda})$ is the *entropic regularizer*. From here, BBVI introduces more structure.

An illustration of the taxonomy is shown in Figure 1. In particular, BBVI has an *infinite sum* structure (IS). That is, it cannot be represented as a sum of finite subcomponents as in ERM. Furthermore,

$$F(\lambda) = \mathbb{E}_{u \sim \varphi} f(\lambda; u) + h(\lambda) \qquad \text{(CP} \cap \text{IS)}$$
$$= \mathbb{E}_{u \sim \varphi} \ell(\mathcal{T}_\lambda(u)) + h(\lambda), \qquad \text{(CP} \cap \text{IS} \cap \text{RP)}$$

where $f(\lambda; u) \triangleq \ell(\mathcal{T}_\lambda(u))$.

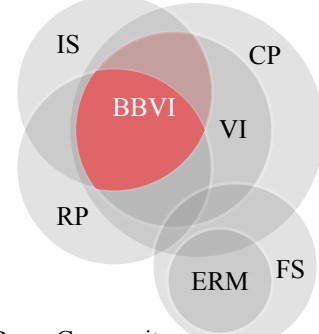

**Theoretical Challenges** The structure of BBVI has multiple challenges that have hindered its theoretical analysis: **(i)** the stochasticity of the Jacobian of $\mathcal{T}$ and **(ii)** The infinite sum structure.

For Item **(i)**, we can see that in

$$\nabla_\lambda \ell(\mathcal{T}_\lambda(u)) = \frac{\partial \mathcal{T}_\lambda(u)}{\partial \lambda} \nabla \ell(\mathcal{T}_\lambda(u)) = \frac{\partial \mathcal{T}_\lambda(u)}{\partial \lambda} g(\lambda; u),$$

| | |
|---|---|
| CP | Composite |
| IS | Infinite Sum |
| RP | Reparameterized |
| FS | Finite Sum |
| ERM | Empirical Risk Minimization |

where $g(\lambda; u) \triangleq (\nabla \ell \circ \mathcal{T}_\lambda)(u)$, both the Jacobian of $\mathcal{T}_\lambda$ and the gradient of the log-likelihood, $g$, depend on the randomness $u$. Effectively decoupling the two is a major challenge to analyzing the properties of the ELBO and its gradient estimators (Domke, 2019, 2020).

For Item **(ii)**, the problem is that recent analyses of SGD (Garrigos & Gower, 2023; Gower *et al.*, 2019; Nguyen *et al.*, 2018; Vaswani *et al.*, 2019) have increasingly been relying on the assumption that $f(\lambda; u)$ is smooth for all $u$ such that

$$\|\nabla_\lambda f(\lambda; u) - \nabla_\lambda f(\lambda'; u)\| \le L\|\lambda - \lambda'\|$$

Figure 1: **Taxonomy of variational inference**. Within BBVI, this work only considers the reparameterization gradient (BBVI $\cap$ RP, shown in **dark red**). This leaves out BBVI with the score gradient (BBVI \ RP, shown in **light red**). The set VI $\cap$ FS includes sparse variational Gaussian processes (Titsias, 2009), while the remaining set VI \ (FS $\cup$ IS $\cup$ RP) includes coordinate ascent VI (Blei *et al.*, 2017).

for some $L < \infty$. This is sensible if the support of $u$ is bounded, which is true for the ERM setting but not for the class of infinite sum (IS) problems. Previous works circumvented this issue by assuming **(i)** that the support of $u$ is bounded (Fujisawa & Sato, 2021) which implicitly changes the variational family, or **(ii)** that the gradient $\nabla f$ is bounded by a constant (Buchholz *et al.*, 2018; Liu & Owen, 2021) which contradicts strong convexity (Nguyen *et al.*, 2018).

# 3 The Evidence Lower Bound Under Nonlinear Scale Parameterizations

Under the linear parameterization ($\phi(x) = x$), the properties of the ELBO, such as smoothness and convexity, have been previously analyzed by Challis & Barber (2013); Domke (2020); Titsias & Lázaro-Gredilla (2014). We generalize these results to nonlinear conditioners.

## 3.1 Technical Assumptions

Let $g_i(\lambda; \boldsymbol{u})$ be the $i$th coordinate of $\boldsymbol{g}(\lambda; \boldsymbol{u})$ and recall that $u_i$ denote the $i$th element of $\boldsymbol{u}$. Establishing convexity and smoothness of the ELBO under nonlinear parameterizations depends on a pair of necessary and sufficient assumptions. To establish smoothness:

**Assumption 3.** The gradient of $\ell$ under reparameterization, g, satisfies
$$|\mathbb{E}g_i(\lambda; \boldsymbol{u})\, u_i \phi''(s_i)| \leq L_s$$
for every coordinate $i = 1, \ldots d$, any $\lambda \in \Lambda$, and some $0 < L_s < \infty$.

Here, $\phi''$ is the second derivative of $\phi$. The next one is required to establish convexity:

**Assumption 4.** The gradient of $\ell$ under reparameterization, g, satisfies
$$\mathbb{E}g_i(\lambda; \boldsymbol{u})\, u_i \geq 0$$
for every coordinate $i = 1, \ldots d$.

Intuitively, these assumption control how much $\nabla \ell$ and $\mathcal{T}_\lambda$ *rotate* the randomness $\boldsymbol{u}$. (Notice that the assumptions are closely related to the matrix $\text{Cov}(g(\lambda; \boldsymbol{u}), \boldsymbol{u})$, the covariance between g and $\boldsymbol{u}$.) However, the peculiar aspect of these assumptions is that they are not implied by the convexity and smoothness of $\ell$. Especially, Assumption 3 strongly depends on the internals of $\nabla \ell$.

## 3.2 Smoothness of the Entropy

Under the linear parameterization, Domke (2020) has previously shown that the entropic regularizer term $h$ is not smooth. This fact immediately implies the ELBO is not smooth. However, certain nonlinear conditioners do result in a smooth regularizer.

**Lemma 1.** *If the diagonal conditioner $\phi$ is $L_h$-log-smooth, then the entropic regularizer $h(\lambda)$ is $L_h$-smooth.*

*Proof.* See the *full proof* in page 24.

**Example 1.** The following diagonal conditioners result in a smooth entropic regularizer:

1. Let $\phi(x) = \text{softplus}(x)$. Then, $h$ is $L_h$-smooth with $L_h \approx 0.167096$.
2. Let $\phi(x) = \exp(x)$. Then, $h$ is $L_h$-smooth for arbitrarily small $L_h$.

This might initially suggest that diagonal conditioners are a promising way of making the ELBO globally smooth. Unfortunately, the properties of the *energy*, $f$, change unfavorably.

## 3.3 Smoothness of the Energy

**Inapplicability of Existing Proof Strategy**   Previously, Domke (2020, Theorem 1) have proven that the energy is smooth when $\phi$ is linear. The key step was to use Bessel's inequality based on the observation that the partial derivatives of the reparameterization function $\mathcal{T}$ form unit bases in expectation. That is,

$$\mathbb{E}\left\langle \frac{\partial \mathcal{T}_\lambda(\boldsymbol{u})}{\partial \lambda_i}, \frac{\partial \mathcal{T}_\lambda(\boldsymbol{u})}{\partial \lambda_j} \right\rangle = \mathbb{1}_{i=j},$$

where $\mathbb{1}_{i=j}$ is an indicator function that is 1 only when $i = j$ and 0 otherwise.

Unfortunately, when $\phi$ is nonlinear, the partial derivatives $\partial \mathcal{T}_\lambda(\boldsymbol{u})/\partial \lambda_i$ for $i = 1, \ldots, p$ no longer form unit bases: while they are still orthogonal in expectation, the *lengths* change nonlinearly depending on $\lambda$. This leaves Bessel's inequality inapplicable. To circumvent this challenge, we establish a replacement for Bessel's inequality:

**Lemma 2.** *Let $\boldsymbol{H}$ be a $n \times n$ symmetric random matrix, where it is bounded as $\|\boldsymbol{H}\|_2 \leq L < \infty$ almost surely. Also, let $\boldsymbol{J}$ be an $m \times n$ random matrix such that $\|\mathbb{E}\boldsymbol{J}^\top \boldsymbol{J}\|_2 < \infty$. Then,*
$$\|\mathbb{E}\boldsymbol{J}^\top \boldsymbol{H}\boldsymbol{J}\|_2 \leq L\|\mathbb{E}\boldsymbol{J}^\top \boldsymbol{J}\|_2.$$

*Proof.* See the *full proof* in page 24.

**Remark 1.** By assuming that the joint log-likelihood $\ell$ is smooth and twice-differentiable, we retrieve Theorem 1 of Domke (2020) by setting $\boldsymbol{J}$ to be the Jacobian of $\mathcal{T}$, and $\boldsymbol{H}$ to be the Hessian of $\ell$ under reparameterization.

**Remark 2.** While our reparameterization function's partial derivatives still form orthogonal bases, they need not be; unlike Bessel's inequality, Lemma 2 does not require this. This implies that Lemma 2 is a strategy more general than Bessel's inequality.

Equipped with Lemma 2, we present our main result on smoothness:

**Theorem 1.** *Let $\ell$ be $L_\ell$-smooth and twice differentiable. Then, the following results hold:*
  *(i) If $\phi$ is linear, the energy $f$ is $L_\ell$-smooth.*
  *(ii) If $\phi$ is 1-Lipschitz, the energy $\ell$ is $(L_\ell + L_s)$-smooth if and only if Assumption 3 holds.*

*Proof.* See the *full proof* in page 27.

Combined with Lemma 1, this directly implies that the overall ELBO is smooth.

**Corollary 1** (Smoothness of the ELBO). *Let $\ell$ be $L_\ell$-smooth and Assumption 3 hold. Furthermore, let the diagonal conditioner be 1-Lipschitz continuous, and $L_\phi$-log-smooth. Then, the ELBO is $(L_\ell + L_s + L_\phi)$-smooth.*

The increase of the smoothness constant implies that we need to use a smaller stepsize to guarantee convergence when using a nonlinear $\phi$. Furthermore, even on simple $L$-smooth examples Assumption 3 may not hold:

**Example 2.** Let $\ell(z) = (1/2) z^\top A z$ and the diagonal conditioner be $\phi(x) = \text{softplus}(x)$. Then,

  **(i)** if $A$ is dense and the variational family is the mean-field family or
  **(ii)** if $A$ is diagonal and the variational family is the Cholesky family,

Assumption 3 holds with $L_s \approx 0.26034 \left(\max_{i=1,\ldots,d} A_{ii}\right)$.

  **(iii)** If $A$ is dense but the Cholesky family is used, Assumption 3 does not hold.

*Proof.* See the *full proof* in page 29.

Example 2 illustrates that establishing the smoothness of the energy becomes non-trivial under nonlinear parameterizations. Even when smoothness does hold, the increased smoothness constant implies that BBVI will be less robust to initialization and stepsizes. Furthermore, in the next section, we will show a much more grave problem: nonlinear parameterizations may affect the convergence *rate*.

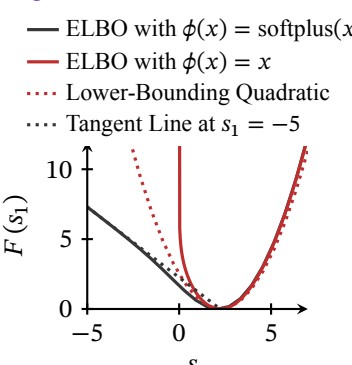

— ELBO with $\phi(x) = \text{softplus}(x)$
— ELBO with $\phi(x) = x$
···· Lower-Bounding Quadratic
···· Tangent Line at $s_1 = -5$

### 3.4 Convexity of the Energy

The convexity of the ELBO under linear parameterizations has first been established by Titsias & Lázaro-Gredilla (2014, Proposition 1) and Domke (2020, Theorem 9). In particular, Domke (2020) show that, when $\phi$ is linear, if $\ell$ is $\mu$-strongly convex, the energy is also $\mu$-strongly convex. However, when using a nonlinear $\phi$ with a co-domain of $\mathbb{R}_+$, which is the whole point of using a nonlinear conditioner, strong convexity of $\ell$ *never* transfers to $f$.

Figure 2: **Optimization landscape resulting from different $\phi$ on a strongly-convex $\ell$.** $\ell$ is the counter-example of Proposition 1 Item **(ii)**. $\phi(x) = x$ preserves strong convexity as shown by the lower-bounding quadratic (red dotted line ····). $\phi = \text{softplus}$ violates the first-order condition of convexity (black dotted line ····).

**Theorem 2.** *Let $\ell$ be $\mu$-strongly convex. Then, we have the following:*
  *(i) If $\phi$ is linear, the energy $f$ is $\mu$-strongly convex.*
  *(ii) If $\phi$ is convex, the energy $f$ is convex if and only if Assumption 4 holds.*
  *(iii) If $\phi$ is such that $\phi \in C^1(\mathbb{R}, \mathbb{R}_+)$, the energy $f$ is not strongly convex.*

*Proof.* See the *full proof* in page 33.
The following proposition provides some conditions for Assumption 4 to hold or not hold.

**Proposition 1.** *We have the following:*
  *(i) If $\ell$ is convex, then for the mean-field family, Assumption 4 holds.*
  *(ii) For the Cholesky family, there exists a convex $\ell$ where Assumption 4 does not hold.*

*Proof.* See the *full proof* in page 31.

For any continuous, differentiable nonlinear conditioner that maps only to non-negative reals, the strong convexity of $\ell$ does lead to a strongly-convex ELBO. This phenomenon is visualized in Figure 2. The loss surface becomes flat near the optimal scale parameter. This problem becomes more noticeable as the optimal scale becomes smaller.

**Nonlinear conditioners are suboptimal.** As the dataset grows, Bayesian posteriors are known to "contract" as characterized by the Bernstein-von Mises theorem (van der Vaart, 1998). That is, the posterior variance becomes close to 0. This behavior also applies to misspecified variational posteriors as shown by Wang & Blei (2019). Thus, for large datasets, nonlinear conditioners mostly operate in the regime where they are suboptimal (locally less strongly convex). But linear conditioners result in a non-smooth entropy (Domke, 2020). This dilemma originally motivated Domke to consider proximal SGD, which we analyze in Section 4.2.

## 4 Convergence Analysis of Black-Box Variational Inference

### 4.1 Black-Box Variational Inference

BBVI with SGD repeats the steps:

$$\lambda_{t+1} = \lambda_t - \gamma_t \left( \widehat{\nabla f}(\lambda_t) + \nabla h(\lambda_t) \right), \quad \text{where} \quad \widehat{\nabla f}(\lambda_t) = \frac{1}{M} \sum_{m=1}^{M} \nabla_\lambda \ell(\mathcal{T}_\lambda(u_m)) \qquad (1)$$

with $u_m \sim \varphi$ is the $M$-sample reparameterization gradient estimator and $\gamma_t$ is the stepsize. (See Kucukelbir et al., 2017 for algorithmic details.)

With our results in Section 3 and the results of Khaled & Richtárik (2023); Kim et al. (2023), we obtain a convergence guarantee. To apply the result of Kim et al. (2023), which bounds the gradient variance, we require an additional assumption.

> **Assumption 5.** The negative log-likelihood $\ell_{\text{like}}(z) \triangleq -\log p(x \mid z)$ is $\mu$-quadratically growing for all $z \in \mathbb{R}^d$ such that
> $$\frac{\mu}{2} \|z - \bar{z}_{\text{like}}\|_2^2 \le \ell_{\text{like}}(z) - \ell_{\text{like}}^*,$$
> where $\bar{z}_{\text{like}}$ is the projection of $z$ to the set of minimizers of $\ell_{\text{like}}$, and $\ell_{\text{like}}^* = \inf_{z \in \mathbb{R}^d} \ell_{\text{like}}(z)$.

This assumption is weaker than assuming that the likelihood satisfies the Polyak-Łojasiewicz inequality (Karimi et al., 2016).

> **Theorem 3.** *Let Assumption 2 hold, the likelihood satisfy Assumption 5, and the assumptions of Corollary 1 hold such that the ELBO $F$ is $L_F$-smooth with $L_F = L_\ell + L_\phi + L_s$. Then, the iterates generated by BBVI through Equation (1) and the $M$-sample reparameterization gradient include an $\epsilon$-stationary point such that $\min_{0 \le t \le T-1} \mathbb{E}\|\nabla F(\lambda_t)\|_2 \le \epsilon$ for any $\epsilon > 0$ if*
> $$T \ge \mathcal{O}\left( \frac{(F(\lambda_0) - F^*)^2 L_F L_\ell^2 C(d, k_\varphi)}{\mu M \epsilon^4} \right)$$
> *for some fixed stepsize $\gamma$, where $C(d, \varphi) = d + k_\varphi$ for the Cholesky family and $C(d, \varphi) = 2k_\varphi\sqrt{d} + 1$ for the mean-field family.*

*Proof.* See the *full proof* in page 35. □

**Remark 3.** Finding an $\epsilon$-stationary point of the ELBO has an iteration complexity of $\mathcal{O}\left(dL_\ell^2 \kappa M^{-1} \epsilon^{-4}\right)$ for the Cholesky family and $\mathcal{O}\left(\sqrt{d}L_\ell^2 \kappa M^{-1} \epsilon^{-4}\right)$ for the mean-field family.

### 4.2 Black-Box Variational Inference with Proximal SGD

**Proximal SGD** For a composite objective $F = f + h$, proximal SGD repeats the steps:

$$\lambda_{t+1} = \text{prox}_{\gamma_t, h}\left(\lambda_t - \gamma_t \widehat{\nabla f}(\lambda_t)\right) = \arg\min_{\lambda \in \Lambda} \left[ \left\langle \widehat{\nabla f}(\lambda_t), \lambda \right\rangle + h(\lambda) + \frac{1}{2\gamma_t}\|\lambda - \lambda_t\|_2^2 \right], \qquad (2)$$

where prox is known as the *proximal* operator and $\gamma_1, \dots, \gamma_T$ is a stepsize schedule.

In the context of VI, proximal SGD has previously been considered by Altosaar et al. (2018); Diao et al. (2023); Khan et al. (2016, 2015). Their overall focus has been on developing alternative algorithms by generalizing $\|\lambda - \lambda^*\|$ to other metrics. In contrast, Domke (2020) considered proximal SGD with the regular Euclidean metric $\|\lambda - \lambda^*\|_2$ for overcoming the non-smoothness of $h$ under

linear parameterizations. Here, we prove the convergence of this scheme and show that it retrieves the fastest known convergence rates in stochastic first-order optimization.

**Proximal Operator for BBVI**   In our context, $h$ is the entropy of $q_\lambda$ in the location-scale family. For this, Domke (2020) show that the the proximal update for $s_1, \dots, s_d$, is

$$\text{prox}_{\gamma_t, h}(s_i) = s_i + \frac{1}{2}\left(\sqrt{s_i^2 + 4\gamma_t} - s_i\right).$$

For other parameters, the proximal operator is the regular gradient descent update in Equation (1).

**Gradient Variance Bound**   We first establish a bound on the gradient variance. In ERM, contemporary strategies do this by exploiting the finite sum structure of the objective (Section 2.4). Here, we establish a variance bound for RP estimator that does not rely on the finite sum assumption.

**Lemma 3** (**Convex Expected Smoothness**). *Let $\ell$ be $L_\ell$-smooth and $\mu$-strongly convex with the variational family satisfying Assumption 2 with the linear parameterization. Then,*

$$\mathbb{E}\|\nabla_\lambda f(\lambda; \mathbf{u}) - \nabla_{\lambda'} f(\lambda'; \mathbf{u})\|_2^2 \leq 2L_\ell \kappa\, C(d, \varphi)\, \text{B}_f(\lambda, \lambda')$$

*holds, where $\text{B}_f(\lambda, \lambda') \triangleq f(\lambda) - f(\lambda') - \langle \nabla f(\lambda'), \lambda - \lambda' \rangle$ is the Bregman divergence, $\kappa = L_\ell/\mu$ is the condition number, $C(d, \varphi) = d + k_\varphi$ for the Cholesky family, and $C(d, \varphi) = 2k_\varphi\sqrt{d} + 1$ for the mean-field family.*

*Proof.* See the *full proof* in page 36.

Furthermore, the gradient variance at the optimum must be bounded:

**Lemma 4** (Domke, 2019; Kim *et al.*, 2023)**.** *Let $\ell$ be $L_\ell$-smooth with the variational family satisfying Assumption 2 and a 1-Lipschitz diagonal conditioner $\phi$. Then, the gradient variance at the optimum $\lambda^* \in \arg\min_{\lambda \in \Lambda} F(\lambda)$ is bounded as*

$$\sigma^2 \leq \frac{1}{M} C(d, \varphi) L_\ell^2 \left(\|\bar{\mathbf{z}} - \mathbf{m}^*\|_2^2 + \|\mathbf{C}^*\|_\text{F}^2\right),$$

*where $\bar{\mathbf{z}}$ is a stationary point of $\ell$, $\mathbf{m}^*$ and $\mathbf{C}^*$ are the location and scale formed by $\lambda^*$, the constants are $C(d, \varphi) = d + k_\varphi$ for the Cholesky family and $C(d, \varphi) = 2k_\varphi\sqrt{d} + 1$ for the mean-field family, $k_\varphi$ is the kurtosis of $\varphi$ as defined in Assumption 1.*

*Proof.* The full-rank case is proven by Domke (2019, Theorem 3), while the mean-field case is a basic corollary of the result by Kim *et al.* (2023, Lemma 2). □

**Remark 4.** The dimensional dependence in the complexity of BBVI is transferred from the variance bound in Lemma 4. Unfortunately, for the Cholesky family, this dimensional dependence in the variance bound is tight (Domke, 2019).

**Main Result**   With the gradient variance bounds, we now present our complexity result. The proof is identical to Theorem 3.2 by Gower *et al.* (2019), where they use a 2-stage decreasing stepsize schedule: the stepsize is initially held constant and then reduced in a $1/t$ rate.

**Theorem 4.** *Let $\ell$ be $L_\ell$-smooth and $\mu$-strongly convex. Then, for any $\epsilon > 0$, BBVI with proximal SGD in Equation (2), the M-sample reparameterization gradient estimator, a variational family satisfying Assumption 2 with the linear parameterization guarantees $\mathbb{E}\|\lambda_T - \lambda^*\|_2^2 \leq \epsilon$ if*

$$\gamma_t = \begin{cases} \frac{M}{2L_\ell \kappa C(d, \varphi)} & \text{for} \quad t \leq 4T_\kappa \\ \frac{2t+1}{(t+1)^2 \mu} & \text{for} \quad t > 4T_\kappa, \end{cases} \qquad T \geq \max\left(\frac{8\sigma^2}{\mu^2 \epsilon} + \frac{4T_\kappa \|\lambda_0 - \lambda^*\|_2}{\text{e}\sqrt{\epsilon}}, \ 4T_\kappa\right)$$

*where $\sigma^2$ is defined in Lemma 4, $T_\kappa = \lceil \kappa^2 C(d, \varphi) M^{-1} \rceil$, $\kappa = L_\ell/\mu$ is the condition number, e is Euler's constant, $\lambda^* = \arg\min_{\lambda \in \Lambda} F(\lambda)$, $C(d, \varphi) = d + k_\varphi$ for the Cholesky family, and $C(d, \varphi) = 2k_\varphi\sqrt{d} + 1$ for the mean-field family.*

*Proof.* See the *full proof* in page 38.

**Remark 5.** BBVI with proximal SGD on $\mu$-strongly convex and $L_\ell$-smooth $\ell$ has a complexity $\mathcal{O}\left(\kappa^2 d M^{-1} \epsilon^{-1}\right)$ for the Cholesky family and $\mathcal{O}\left(\kappa^2 \sqrt{d} M^{-1} \epsilon^{-1}\right)$ for the mean-field family.

**Remark 6.** We also provide a similar result with a fixed stepsize in Theorem 7 of Appendix F.3.2. In this case, the complexity is $\mathcal{O}\left(\kappa^2 d M^{-1} \epsilon^{-1} \log \epsilon^{-1}\right)$ for the Cholesky family and $\mathcal{O}\left(\kappa^2 \sqrt{d} M^{-1} \epsilon^{-1} \log \epsilon^{-1}\right)$ for the mean-field family.

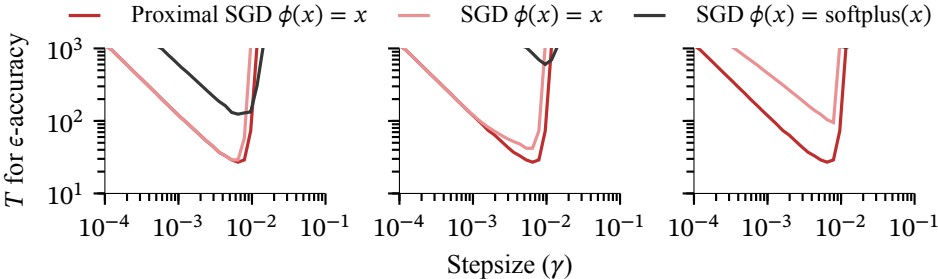

Figure 3: **Stepsize versus the number of iterations for vanilla SGD and proximal SGD to achieve** $D_{\mathrm{KL}}(q_\lambda, \pi) \leq \epsilon = 1$ **under different initializations for Gaussian posteriors.** The initializations $C(\lambda_0)$ are $\mathbf{I}$, $10^{-3}\mathbf{I}$, $10^{-5}\mathbf{I}$ from left to right, respectively. The average suboptimality at iteration $t$ was estimated from 10 independent runs. For each run, the target posterior was a 10-dimensional Gaussian with a covariance with a condition number $\kappa = 10$ and a smoothness of $L = 100$.

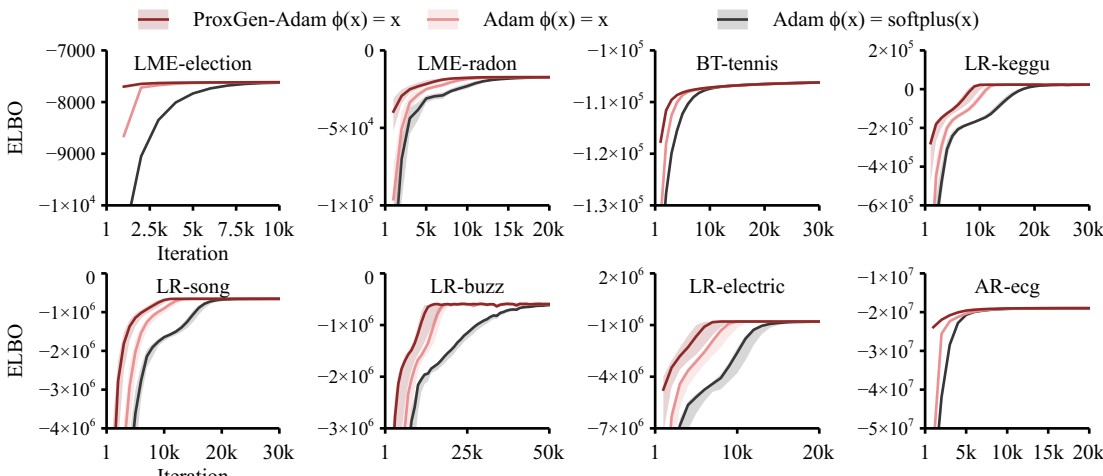

Figure 4: **Comparison of BBVI convergence speed (ELBO v.s. Iteration) of different optimization algorithms.** The error bands are the 80% quantiles estimated from 20 (10 for AR-eeg) independent replications. The results shown used a base stepsize of $\gamma = 10^{-3}$, while the initial point was $\boldsymbol{m}_0 = \mathbf{0}, \boldsymbol{C}_0 = \mathbf{I}$. Details on the setup can be found in the text of Section 5.2 and Appendix G.

## 5 Experiments

### 5.1 Synthetic Problem

**Setup** We first compare proximal SGD against vanilla SGD with linear and nonlinear parameterizations on a synthetic problem, which is log-smooth, strongly log-concave, and the exact solution is known. While a similar experiment was already conducted by Domke (2020), here we include nonlinear parameterizations, which were not originally considered. We run all algorithms with a fixed stepsize to infer a multivariate Gaussian with a full-rank covariance matrix. The variational approximation is a full-rank Gaussian formed by $\varphi = \mathcal{N}(0, 1)$ and the Cholesky parameterization.

**Results** The results are shown in Figure 3. Proximal SGD is clearly the most robust against initialization. Also, SGD with the nonlinear parameterization $\phi(x) = \mathrm{softplus}(x)$ is much slower to converge under all initializations. This confirms that linear parameterizations are indeed superior for both robustness against initializations and convergence speed.

### 5.2 Realistic Problems

**Setup** We now evaluate proximal SGD on realistic problems. In practice, Adam (Kingma & Ba, 2015) is observed to be robust against stepsize choices (Zhang et al., 2019). The reason why Adam performs well on non-smooth, non-convex problems is still under investigation (Kunstner et al., 2023; Reddi et al., 2023; Zhang et al., 2022). Nonetheless, to compare fairly against Adam, we implement a recently proposed variant of proximal SGD called ProxGen (Yun et al., 2021), which

includes an Adam-like update rule. The probabilitic models and datasets are fully described in [Appendix G](). We implement these models and BBVI on top of the Turing ([Ge et al., 2018]()) probabilistic programming framework. Due to the size of these datasets, we implement *doubly stochastic* subsampling ([Titsias & Lázaro-Gredilla, 2014]()) with a batch size of $B = 100$ ($B = 500$ for BT-tennis) with $M = 10$ Monte Carlo samples. For batch subsampling, we implement random-reshuffling, which is faster than independent subsampling both empirically ([Bottou, 2009]()) and theoretically ([Ahn et al., 2020]()[; Haochen & Sra, 2019]()[; Mishchenko et al., 2020]()[; Nagaraj et al., 2019]()). We also observe that doubly stochastic BBVI benefits from reshuffling, but leave a detailed investigation to future works.

**Results**  Representative results are shown in [Figure 4](), with additional results in [Appendix H](). Both ProxGen-Adam and Adam with linear parameterizations converge faster than Adam with nonlinear parameterization. Furthermore, for the case of election and buzz, Adam with the nonlinear parameterization converges much slower than the alternatives. When using linear parameterizations, ProxGen-Adam appears to be generally faster than Adam. We note, however, that due to the difference in the update rule between ProxGen-Adam and Adam, proximal operators alone might not fully explain the performance difference. Nevertheless, the results of our experiment do conclusively suggest that linear parameterizations are superior.

## 6  Discussions

**Conclusions**  In this work, we have proven the convergence of BBVI. Our assumptions encompass implementations that are actually used in practice, and our theoretical analysis revealed limitations in some of the popular design choices (mainly the use of nonlinear conditioners). To resolve this issue, we re-evaluated the utility of proximal SGD both theoretically and practically, where it achieved the strongest theoretical guarantees in stochastic first-order optimization.

**Related Works**  To prove the convergence of BBVI, early works have *a-priori* "assumed" the regularity of the ELBO and the gradient estimator ([Alquier & Ridgway, 2020]()[; Buchholz et al., 2018]()[; Khan et al., 2016]()[, 2015]()[; Liu & Owen, 2021]()[; Regier et al., 2017]()). Towards a more rigorous understanding, [Domke (2019)](); [Fan et al. (2015)](); [Kim et al. (2023)](); [Xu et al. (2019)]() studied the reparameterization gradient, [Xu & Campbell (2022)]() studied the asymptotics of the ELBO, [Challis & Barber (2013)](); [Domke (2020)](); [Titsias & Lázaro-Gredilla (2014)]() established convexity, and [Domke (2020)]() established smoothness. On the other hand, [Bhatia et al. (2022)](); [Hoffman & Ma (2020)]() established rigorous convergence guarantees by considering simplified variant of BBVI where only the scale is optimized, and [Fujisawa & Sato (2021)]() assumed that the support of $\varphi$ is bounded almost surely. Meanwhile, under similar assumptions to ours, [Diao et al. (2023)](); [Lambert et al. (2022)]() recently established convergence guarantees for proximal SGD BBVI with a Bures-Wasserstein metric. Their computational properties differ from BBVI as they require Hessian evaluations. Also, understanding BBVI, which is VI with a Euclidean metric, is an important problem due to its practical relevance.

**Limitations**  Our work has multiple limitations: **(i)** Our results are restricted to the location-scale family, **(ii)** the reparameterization gradient, and **(iii)** smooth joint log-likelihoods. However, the location-scale family with the reparameterization gradient is the most widely used combination in practice, and replacing the smoothness assumption is an active area of research in stochastic optimization. For our results on proximal SGD, we further assume that the joint log-likelihood is $\mu$-strongly convex (equivalently strongly log-concave posteriors). It is unclear how to extend the guarantees to only smooth but non-log-concave joint log-likelihoods.

**Open Problems**  Although we have proven that the mean-field dimensional family has a dimension dependence of $\mathcal{O}\left(\sqrt{d}\right)$, empirical results suggest room for improvement ([Kim et al., 2023]()). Therefore, we pose the following conjecture:

**Conjecture 1.** *Under mild assumptions, BBVI for the mean-field variational family converges with only logarithmic dimensional dependence or no explicit dimensional dependence at all.*

This would put mean-field BBVI in a regime clearly faster than approximate MCMC ([Freund et al., 2022]()). Also, it is unknown whether the $\mathcal{O}\left(\kappa^2\right)$ condition number dependence dependence is tight. In fact, for proximal SGD BBVI in Bures-Wasserstien space, [Diao et al. (2023)]() report a dependence of $\mathcal{O}\left(\kappa\right)$. Lastly, it would be interesting to see whether natural gradient VI (NGVI; [Amari, 1998]()[; Khan & Lin, 2017]()) can achieve similar convergence guarantees. While it is empirically known that NGVI often converges faster ([Lin et al., 2019]()), theoretical evidence has yet to follow.

## Acknowledgments and Disclosure of Funding

The authors would like to thank Justin Domke for discussions on the concurrent results, Javier Burroni for pointing out a mistake in the earlier version of this work, and the anonymous reviewers for their constructive comments.

K. Kim and J. R. Gardner were funded by the National Science Foundation Award [IIS-2145644], while Y.-A. Ma was funded by the National Science Foundation Grants [NSF-SCALE MoDL-2134209] and [NSF-CCF-2112665 (TILOS)], the U.S. Department Of Energy, Office of Science, and the Facebook Research award.

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

# On the Convergence of
# Black-Box Variational Inference
## *Appendix*

---

**Table of Contents**

# A  Computational Resources

Table 1: Computational Resources

| Type | Model and Specifications |
|---|---|
| System Topology | 2 nodes with 2 sockets each with 24 logical threads (total 48 threads) |
| Processor | 1 Intel Xeon Silver 4310, 2.1 GHz (maximum 3.3 GHz) per socket |
| Cache | 1.1 MiB L1, 30 MiB L2, and 36 MiB L3 |
| Memory | 250 GiB RAM |
| Accelerator | 1 NVIDIA RTX A5000 per node, 2 GHZ, 24GB RAM |

Running the experiments took approximately a week.

# B  Nomenclature

| Symbol | Definition | Description | Section |
|---|---|---|---|
| $\lambda$ | | Variational parameters | 2.1 |
| $z$ | | Parameters of the target model $\pi$ | 2.1 |
| $\mathcal{T}_\lambda$ | $\triangleq C(\lambda)\,u + m$ | location-scale reparameterization function | 2.2 |
| $u$ | | Random vector before reparameterization | 2.2 |
| $\varphi$ | | Base distribution of $u$ | 2.2 |
| $m$ | | Location parameter (part of $\lambda$) | 2.2 |
| $C$ | | Scale parameter (part of $\lambda$) | 2.2 and 2.3 |
| $\phi$ | | Diagonal conditioner | 2.3 |
| $k_\varphi$ | | Kurtosis (non-central 4th moment) of $u$ | 2.3 |
| $D_\phi(s)$ | | Diagonal of $C$ using the diagonal conditioner $\phi$ | 2.3 |
| $L$ | | Strictly lower triangular part of $C$ | 2.3 |
| $s$ | | Elements forming the diagonal of $C$ | 2.3 |
| $\ell(z)$ | $\triangleq -\log p(z, x)$ | Negative joint likelihood | 2.4 |
| $f(\lambda)$ | $\triangleq \mathbb{E}_{z \sim q_\lambda}\ell(z)$ | Energy | 2.4 |
| $h(\lambda)$ | $\triangleq -\mathbb{H}(q_\lambda)$ | Negative entropy | 2.4 |
| $F(\lambda)$ | $\triangleq f(\lambda) + h(\lambda)$ | Negative ELBO | 2.1 and 2.4 |
| $f(\lambda; u)$ | $\triangleq \ell(\mathcal{T}_\lambda(u))$ | Negative Log-Likelihood under reparameterization | 2.4 |
| $g(\lambda; u)$ | $\triangleq \nabla\ell(\mathcal{T}_\lambda(u))$ | Gradient of the Log-likelihood under reparameterization | 2.4 |
| $M$ | | Number of Monte Carlo samples | 2.1 |
| $\gamma_t$ | | Stepsize of (proximal) SGD at iteration $t$ | 4.1 and 4.2 |
| $\widehat{\nabla f}$ | | Reparameterization gradient estimator of the energy | 4.1 |

# C  Definitions

For completeness, we provide formal definitions for some of the terms we used throughout the paper.

**Definition 5 (Smoothness).** A function $f : \mathcal{Z} \to \mathbb{R}$ is said to be *L*-smooth if the inequality

$$\|\nabla f(z) - \nabla f(z')\| \leq \|z - z'\|$$

holds for all $z, z' \in \mathcal{Z}$.

This assumption, also occasionally called Lipschitz smoothness, restricts the amount the gradient can change for a given distance. When $f$ is twice differentiable, an equivalent condition is the Hessian to be bounded:

**Definition 6 (Smoothness).** A twice differentiable function $f : \mathcal{Z} \to \mathbb{R}$ is said to be *L*-smooth if the inequality

$$\|\nabla^2 f(z)\| \leq L$$

holds for all $z \in \mathcal{Z}$.

**Remark 7.** Assuming a function $f$ is smooth is equivalent to assuming that $f$ can be upper bounded by a quadratic function everywhere.

**Remark 8.** When the log-density $\log \pi$ of a probability measure $\Pi$ is *L*-smooth, $\log \pi$ can be upper bounded everywhere by the log-density of a Gaussian.

**Definition 7 (Strong Convexity).** A twice differentiable function $f : \mathbb{R}^d \to \mathbb{R}$ is said to be $\mu$-strongly convex if the inequality

$$\frac{\mu}{2}\|z - z'\|^2 + \langle \nabla f(z), z - z' \rangle + f(z) \leq f(z')$$

holds for all $z, z' \in \mathbb{R}^d$ and some $\mu > 0$.

**Remark 9.** If Definition 7 holds only for $\mu = 0$, $f$ is said to be (non-strongly) convex.

**Remark 10.** Assuming a function $f$ is strongly convex is equivalent to assuming that $f$ can be lower bounded by a quadratic.

**Definition 8 (Strongly Log-Concave Measures).** For a probability measure $\Pi$ in a Euclidean measurable space $(\mathbb{R}^d, \mathcal{B}(\mathbb{R}^d), \mathbb{P})$, where $\mathcal{B}(\mathbb{R}^d)$ is the $\sigma$-algebra of Borel-measurable subsets of $\mathbb{R}^d$, $\mathbb{P}$ is the Lebesgue measure, we say $\Pi$ is $\mu$-strongly log-concave if its log-density $\log \pi(z) : \mathbb{R}^d \to \mathbb{R}$ is $\mu$-strongly convex for some $\mu > 0$.

**Remark 11.** If Definition 8 holds only for $\mu = 0$, $\Pi$ is said to be (non-strongly) log-concave.

**Remark 12.** When $\Pi$ is $\mu$-strongly log-concave, $\log \pi$ can be lower bounded everywhere by the log-density of a Gaussian.

# D    ProxGen Adam for Black-Box Variational Inference

---

**Algorithm 1:** ProxGen-Adam for Black-Box Variational Inference

---

**Input:** Initial variational parameters $\boldsymbol{\lambda}_0$, base stepsize $\alpha$, second moment stepsize $\beta_2$,
  momentum stepsize $\{\beta_{1,t}\}_{t=1}^T$, small positve constant $\epsilon$

**for** $t = 1, \dots, T$ **do**

    estimate gradient of energy $\widehat{\nabla}f$

    $\boldsymbol{g}_t = \widehat{\nabla}f(\boldsymbol{\lambda}) + \nabla h(\boldsymbol{\lambda})$

    $\overline{\boldsymbol{\lambda}}_{t+1} = \beta_{1,t}\overline{\boldsymbol{\lambda}}_t + (1 - \beta_{1,t})\overline{\boldsymbol{\lambda}}_t$

    $\boldsymbol{v}_{t+1} = \beta_2\boldsymbol{v}_t + (1 - \beta_2)\boldsymbol{g}_t^2$

    $\boldsymbol{\Gamma}_{t+1} = \operatorname{diag}\left(\alpha/\left(\sqrt{\boldsymbol{v}_{t+1}} + \epsilon\right)\right)$

    $\boldsymbol{\lambda}_{t+1} = \boldsymbol{\lambda}_t - \boldsymbol{\Gamma}_{t+1}\overline{\boldsymbol{\lambda}}_{t+1}$

    $\boldsymbol{s}_{t+1} \leftarrow \operatorname{getscale}(\boldsymbol{\lambda}_{t+1})$

    $\boldsymbol{s}_{t+1} \leftarrow \boldsymbol{s}_{t+1} + \frac{1}{2}\left(\sqrt{\boldsymbol{s}_{t+1}^2 + 4\boldsymbol{\gamma}_{s,t+1}} - \boldsymbol{s}_{t+1}\right)$

    $\boldsymbol{\lambda}_{t+1} \leftarrow \operatorname{setscale}(\boldsymbol{\lambda}_{t+1}, \boldsymbol{s}_{t+1})$

**end**

---

(By convention, all vector operations are elementwise.)

Adaptive and matrix-valued stepsize-variants of SGD such as Adam (Kingma & Ba, 2015), Ada-Grad (Duchi *et al.*, 2011) are widely used. The matrix stepsize of Adam at iteration $t$ is given as

$$\boldsymbol{\Gamma}_{t+1} = \operatorname{diag}\left(\alpha/\left(\sqrt{\boldsymbol{v}_{t+1}} + \epsilon\right)\right),$$

where $\boldsymbol{v}_t$ is the exponential moving average of the second moment, $\alpha$ is the "base stepsize." Furthermore, the matrix stepsize is applied to the moving average of the gradients, a scheme often called the (heavy-ball) momentum, denoted here as $\overline{\boldsymbol{\lambda}}_t$.

Recently, Yun *et al.* (2021) have proven the convergence for these adaptive, momentum, and matrix-valued stepsize-based SGD methods with proximal steps. Then, the proximal operator is applied as

$$\operatorname{prox}_{\boldsymbol{\Gamma}_t,h}\left(\boldsymbol{\lambda}_t - \boldsymbol{\Gamma}_t\overline{\boldsymbol{\lambda}}_t\right) = \arg\min_{\boldsymbol{\lambda}} \left\{\langle\overline{\boldsymbol{\lambda}}_t, \boldsymbol{\lambda}\rangle + h(\boldsymbol{\lambda}) + \frac{1}{2}(\boldsymbol{\lambda} - \boldsymbol{\lambda}_t)^\top\boldsymbol{\Gamma}_t^{-1}(\boldsymbol{\lambda} - \boldsymbol{\lambda}_t)\right\}.$$

For Adam, the matrix-valued stepsize is a diagonal matrix. Thus, the proximal operator of Domke (2020) for each $s_i$ forms independent 1-dimensional quadratic problems. Thus, the proximal step is given in the closed-form

$$\operatorname{prox}_{\boldsymbol{\Gamma}_t,h}(s_i) = s_i + \frac{1}{2}\left(\sqrt{s_i^2 + 4\gamma_{s_i}} - s_i\right),$$

where, dropping the index $t$ for clarity, $\bar{s}_i$ is the element of $\overline{\boldsymbol{\lambda}}_t$ corresponding to $s_i$, $\gamma_{s_i}$ denotes the stepsize of $s_i$ (a diagonal element of $\boldsymbol{\Gamma}_t$). Combined with the Adam-like stepsize rule, the algorithm is shown in Algorithm 1.

**Difference with Adam**    In Algorithm 1, we can see the differences with vanilla Adam. Notably, ProxGenAdam does not perform bias correction of the estimated moments. Furthermore, while some implementations of Adam decay $\beta_1$, we keep it constant. It is possible that these differences could result in a different behavior from vanilla Adam. However, in this work, we follow the original implementation by Yun *et al.* (2021) as closely as possible and leave the comparison with vanilla Adam to future works.

# E   Detailed Comparison Against Domke *et al.* (2023)

In this section, we contrast our results against those of Domke *et al.* (2023). First, the main challenge to establishing a convergence guarantee for BBVI has been on bounding the gradient variance. In particular, Domke (2019) proved that the variance of the reparameterization gradient for the energy, $\widehat{\nabla f}$, is bounded as

$$\mathbb{E}\|\widehat{\nabla f}(\lambda)\|_2^2 \leq \alpha\|\lambda - \bar{\lambda}\|_2^2 + \beta \tag{3}$$

for some finite positive constants $\alpha, \beta$ depending on the problem constants $d, L, k_\varphi$. Domke *et al.* (2023) call a gradient estimator satisfying this bound to be a "quadratic variance" estimator. Furthermore, they prove that the closed-form entropy (CFE; Kucukelbir *et al.*, 2017; Titsias & Lázaro-Gredilla, 2014) estimator:

$$\widehat{\nabla F}_{\mathrm{CFE}}(\lambda) \triangleq \widehat{\nabla f}(\lambda) + \nabla h(\lambda)$$

and the STL estimator by Roeder *et al.* (2017):

$$\widehat{\nabla F}_{\mathrm{STL}}(\lambda) \triangleq \frac{1}{M}\sum_{m=1}^{M} -\nabla_\lambda \ell\left(\mathcal{T}_\lambda(\boldsymbol{u}_m)\right) + \nabla_\lambda \log q_\lambda\left(\mathcal{T}_\nu(\boldsymbol{u}_m)\right)\Big|_{\nu=\lambda},$$

where $\boldsymbol{u} \sim \varphi$, also qualify as quadratic variance estimators.

Unfortunately, it has been unknown whether SGD is guaranteed to converge with a quadratic variance estimator except for strongly convex objectives (Wright & Recht, 2021, p. 85). Domke *et al.* (2023) expand the boundaries of SGD and prove that projected and proximal SGD with a quadratic variance estimator converges for both convex and strongly convex objectives. In particular, for the location-scale variational family, the linear parameterization, and log-concave objectives, they prove a complexity of $\mathcal{O}\left(1/\varepsilon^2\right)$, and for strongly log-concave objectives, they prove a complexity of $\mathcal{O}\left(1/\varepsilon\right)$.

On the other hand, Kim *et al.* (2023) and Lemma 12 developed the bound in Equation (3) to be of the form of

$$\mathbb{E}\|\widehat{\nabla F}(\lambda)\|_2^2 \leq A\left(F(\lambda) - F^*\right) + \|\nabla F(\lambda)\|_2^2 + C \tag{4}$$

for some positive finite constants $A, B, C$, for which the convergence of SGD for convex, strongly convex (Garrigos & Gower, 2023; Gorbunov *et al.*, 2020), and non-convex objectives (Khaled & Richtárik, 2023) have already been proven. Applying these results to log-smooth and log-quadratically growing objectives, we prove a complexity of $\mathcal{O}\left(1/\varepsilon^4\right)$, while for strong log-concave objectives, we also prove a complexity of $\mathcal{O}\left(1/\varepsilon\right)$.

Overall, both approaches can be summarized as follows: we focused on establishing gradient variance bounds of known convergence proofs, while Domke *et al.* (2023) aimed to prove that the bound by Domke (2019) is sufficient to guarantee convergence. Note that, for strongly log-concave objectives, Equation (3) immediately implies Equation (4). Therefore, both approaches intersect in the case of strongly log-concave objectives. Indeed, Theorem 8 and the analogous result of Domke *et al.* (2023) are both based on the same proof strategy by Gower *et al.* (2019).

# F  Proofs

## F.1  Auxiliary Lemmas

**Lemma 5.** *Let $\phi(x) = x$. Then, the parameterization is linear in the sense that $\mathcal{T}_\lambda$ is a bilinear function such that*

$$\mathcal{T}_{\lambda-\lambda'}(\boldsymbol{u}) = \mathcal{T}_\lambda(\boldsymbol{u}) - \mathcal{T}_{\lambda'}(\boldsymbol{u}).$$

*for any $\lambda, \lambda' \in \Lambda$.*

*Proof.*

$$\mathcal{T}_{\lambda-\lambda'}(\boldsymbol{u}) = (\boldsymbol{C}(\lambda-\lambda'))\boldsymbol{u} + (\boldsymbol{m}-\boldsymbol{m}')$$
$$= (\boldsymbol{D}_\phi(\boldsymbol{s}-\boldsymbol{s}') + (\boldsymbol{L}-\boldsymbol{L}'))\boldsymbol{u} + (\boldsymbol{m}-\boldsymbol{m}'),$$

using the fact that $\phi$ is the identity function,

$$= (\boldsymbol{D}_\phi(\boldsymbol{s}) - \boldsymbol{D}_\phi(\boldsymbol{s}') + (\boldsymbol{L}-\boldsymbol{L}'))\boldsymbol{u} + (\boldsymbol{m}-\boldsymbol{m}')$$
$$= (\boldsymbol{C}(\lambda) - \boldsymbol{C}(\lambda'))\boldsymbol{u} + (\boldsymbol{m}+\boldsymbol{m}')$$
$$= (\boldsymbol{C}(\lambda)\boldsymbol{u} + \boldsymbol{m}) - (\boldsymbol{C}(\lambda')\boldsymbol{u} + \boldsymbol{m}')$$
$$= \mathcal{T}_\lambda(\boldsymbol{u}) - \mathcal{T}_{\lambda'}(\boldsymbol{u}).$$

The linearity with respect to $\boldsymbol{u}$ is obvious. $\square$

**Lemma 6.** *Let the linear parameterization be used. Then, for any $\lambda, \lambda' \in \Lambda$, the inner product of the Jacobian of the reparameterization function satisfies the following equalities for any $\boldsymbol{u} \in \mathbb{R}^d$.*

*(i)  For the Cholesky family (Domke, 2019, Lemma 8),*

$$\left(\frac{\partial \mathcal{T}_\lambda(\boldsymbol{u})}{\partial \lambda}\right)^\top \frac{\partial \mathcal{T}_\lambda(\boldsymbol{u})}{\partial \lambda} = \left(1 + \|\boldsymbol{u}\|_2^2\right)\mathbf{I}$$

*(ii)  For the mean-field family (Kim et al., 2023, Lemma 1),*

$$\left(\frac{\partial \mathcal{T}_\lambda(\boldsymbol{u})}{\partial \lambda}\right)^\top \frac{\partial \mathcal{T}_\lambda(\boldsymbol{u})}{\partial \lambda} = \left(1 + \|\boldsymbol{U}^2\|_{\mathrm{F}}\right)\mathbf{I},$$

*where $\boldsymbol{U} = \mathrm{diag}(u_1, \dots, u_d)$.*

**Lemma 7.** *Let the linear parameterization be used. Then, for any $\lambda \in \Lambda$ and any $\boldsymbol{z} \in \mathbb{R}^d$, the following relationships hold.*

*(i)  For the Cholesky family (Domke, 2019, Lemma 2),*

$$\mathbb{E}\left(1 + \|\boldsymbol{u}\|_2^2\right)\|\mathcal{T}_\lambda(\boldsymbol{u}) - \boldsymbol{z}\|_2^2 = (d+1)\|\boldsymbol{m} - \boldsymbol{z}\|_2^2 + \left(d + k_\varphi\right)\|\boldsymbol{C}\|_{\mathrm{F}}^2$$

*(ii)  For the mean-field family (Kim et al., 2023, Lemma 2),*

$$\mathbb{E}\left(1 + \|\boldsymbol{U}^2\|_{\mathrm{F}}\right)\|\mathcal{T}_\lambda(\boldsymbol{u}) - \boldsymbol{z}\|_2^2 \le \left(\sqrt{dk_\varphi} + k_\varphi\sqrt{d} + 1\right)\|\boldsymbol{m} - \boldsymbol{z}\|_2^2 + \left(2k_\varphi\sqrt{d} + 1\right)\|\boldsymbol{C}\|_{\mathrm{F}}^2.$$

**Corollary 2.** *Let the linear parameterization be used and $\lambda, \lambda' \in \Lambda$ be any pair of variational parameters.*

*(i)  For the Cholesky family,*

$$\mathbb{E}\left(1 + \|\boldsymbol{u}\|_2^2\right)\|\mathcal{T}_{\lambda'}(\boldsymbol{u}) - \mathcal{T}_\lambda(\boldsymbol{u})\|_2^2 \le (k_\varphi + d)\|\lambda - \lambda'\|_2^2$$

*(ii)  For the mean-field family,*

$$\mathbb{E}\left(1 + \|\boldsymbol{U}^2\|_{\mathrm{F}}\right)\|\mathcal{T}_{\lambda'}(\boldsymbol{u}) - \mathcal{T}_\lambda(\boldsymbol{u})\|_2^2 \le (2k_\varphi\sqrt{d} + 1)\|\lambda - \lambda'\|_2^2$$

*Proof.*  The results are a direct consequence of Lemma 7 and Lemma 5.

**Proof of (i)**  We start from Lemma 7 as

$$\mathbb{E}\left(1 + \|\boldsymbol{u}\|_2^2\right)\|\mathcal{T}_{\lambda-\lambda'}\left(\boldsymbol{u}\right) - \boldsymbol{z}\|_2^2 = (d+1)\|(\boldsymbol{m}-\boldsymbol{m}') - \boldsymbol{z}\|_2^2 + (d+k_\varphi)\|\boldsymbol{C}\left(\lambda\right) - \boldsymbol{C}\left(\lambda'\right)\|_F^2,$$

setting $\boldsymbol{z} = \boldsymbol{0}$,

$$= (d+1)\|\boldsymbol{m}-\boldsymbol{m}'\|_2^2 + (d+k_\varphi)\|\boldsymbol{C}\left(\lambda\right) - \boldsymbol{C}\left(\lambda'\right)\|_F^2,$$

and since $k_\varphi \geq 3$ by the property of the kurtosis,

$$\leq \left(d + k_\varphi\right)\left(\|\boldsymbol{m}-\boldsymbol{m}'\|_2^2 + \|\boldsymbol{C}\left(\lambda\right) - \boldsymbol{C}\left(\lambda'\right)\|_F^2\right)$$

$$= \left(d + k_\varphi\right)\|\lambda - \lambda'\|_2^2.$$

**Proof of (ii)**  Similarly, for the mean-field family, we can apply Lemma 7 as

$$\mathbb{E}\left(1 + \|\boldsymbol{U}^2\|_F\right)\|\mathcal{T}_{\lambda-\lambda'}\left(\boldsymbol{u}\right) - \bar{\boldsymbol{z}}\|_2^2 \leq \left(\sqrt{dk_\varphi} + k_\varphi\sqrt{d} + 1\right)\|(\boldsymbol{m}-\boldsymbol{m}') - \bar{\boldsymbol{z}}\|_2^2 + \left(2k_\varphi\sqrt{d} + 1\right)\|\boldsymbol{C} - \boldsymbol{C}'\|_F^2,$$

setting $\boldsymbol{z} = \boldsymbol{0}$,

$$= \left(\sqrt{dk_\varphi} + k_\varphi\sqrt{d} + 1\right)\|\boldsymbol{m}-\boldsymbol{m}'\|_2^2 + \left(2k_\varphi\sqrt{d} + 1\right)\|\boldsymbol{C} - \boldsymbol{C}'\|_F^2,$$

and since $k_\varphi \geq 3$ by the property of the kurtosis,

$$\leq \left(2k_\varphi\sqrt{d} + 1\right)\left(\|\boldsymbol{m}-\boldsymbol{m}'\|_2^2 + \|\boldsymbol{C}\left(\lambda\right) - \boldsymbol{C}\left(\lambda'\right)\|_F^2\right)$$

$$= \left(2k_\varphi\sqrt{d} + 1\right)\|\lambda - \lambda'\|_2^2.$$

$\square$

**Lemma 8.** *For the linear parameterization,*

$$\mathbb{E}\|\mathcal{T}_\lambda\left(\boldsymbol{u}\right) - \mathcal{T}_{\lambda'}\left(\boldsymbol{u}\right)\|_2^2 = \|\lambda - \lambda'\|_2^2$$

*for any $\lambda, \lambda' \in \Lambda$.*

*Proof.*  First notice that, for linear parameterizations, we have

$$\mathbb{E}\|\mathcal{T}_\lambda\left(\boldsymbol{u}\right)\|_2^2 = \mathbb{E}\|\boldsymbol{Cu} + \boldsymbol{m}\|_2^2$$

$$= \mathbb{E}\boldsymbol{u}^\top\boldsymbol{C}^\top\boldsymbol{Cu} + \|\boldsymbol{m}\|_2^2 + 2\boldsymbol{m}^\top\boldsymbol{C}\mathbb{E}\boldsymbol{u}$$

$$= \mathbb{E}\operatorname{tr}\left(\boldsymbol{u}^\top\boldsymbol{C}^\top\boldsymbol{Cu}\right) + \|\boldsymbol{m}\|_2^2 + 2\boldsymbol{m}^\top\boldsymbol{C}\mathbb{E}\boldsymbol{u},$$

rotating the elements of the trace,

$$= \operatorname{tr}\left(\boldsymbol{C}^\top\boldsymbol{C}\mathbb{E}\boldsymbol{u}\boldsymbol{u}^\top\right) + \|\boldsymbol{m}\|_2^2 + 2\boldsymbol{m}^\top\boldsymbol{C}\mathbb{E}\boldsymbol{u},$$

applying Assumption 1

$$= \operatorname{tr}\left(\boldsymbol{C}^\top\boldsymbol{C}\right) + \|\boldsymbol{m}\|_2^2$$

$$= \|\boldsymbol{C}\|_F^2 + \|\boldsymbol{m}\|_2^2$$

$$= \|\lambda\|_2^2.$$

Combined with Lemma 5, we have

$$\mathbb{E}\|\mathcal{T}_\lambda\left(\boldsymbol{u}\right) - \mathcal{T}_{\lambda'}\left(\boldsymbol{u}\right)\|_2^2 = \mathbb{E}\|\mathcal{T}_{\lambda-\lambda'}\left(\boldsymbol{u}\right)\|_2^2 = \|\lambda - \lambda'\|_2^2.$$

$\square$

### F.2 Properties of the Evidence Lower Bound

#### F.2.1 Smoothness

**Lemma 1.** *If the diagonal conditioner $\phi$ is $L_h$-log-smooth, then the entropic regularizer $h(\lambda)$ is $L_h$-smooth.*

*Proof.* The entropic regularizer is

$$h(\lambda) = -\mathrm{H}(\varphi) - \sum_{i=1}^{d} \log \phi(s_i),$$

and depends only on the diagonal elements $s_1, \ldots, s_d$ of $C$. The Hessian of $h$ is then a diagonal matrix, where only the entries that correspond to $s_1, \ldots, s_d$ are non-zero. The Lipschitz smoothness constant is then the constant $L_h < \infty$ that satisfies

$$\frac{\partial^2 h(\lambda)}{\partial s_i^2} = -\frac{d^2 \log \phi}{ds_i^2} < L_h$$

for all $i = 1, \ldots, d$, which is the smoothness constant of $s_i \mapsto \log \phi(s_i)$. $\qquad\square$

**Lemma 2.** *Let $H$ be a $n \times n$ symmetric random matrix, where it is bounded as $\|H\|_2 \leq L < \infty$ almost surely. Also, let $J$ be an $m \times n$ random matrix such that $\|\mathbb{E}J^\top J\|_2 < \infty$. Then,*
$$\|\mathbb{E}J^\top H J\|_2 \leq L \|\mathbb{E}J^\top J\|_2.$$

*Proof.* By the property of the Rayleigh quotients, for a symmetric matrix $A$, its maximum eigenvalue is given in the variational form

$$\sup_{\|x\| \leq 1} x^\top H x = \sigma_{\max}(H) \leq \sqrt{\sigma_{\max}(H)^2} = \|H\|_2,$$

where $\sigma_{\max}(A)$ is the maximal eigenvalue of $A$. Notice the relationship with the $\ell_2$-operator norm. The inequality is strict only if all eigenvalues are negative.

From the property above,

$$\|\mathbb{E}J^\top H J\|_2 = \sup_{\|x\|_2 \leq 1} x^\top \left(\mathbb{E}J^\top H J\right) x.$$

By reparameterizing as $y = Jx$,

$$= \sup_{\|x\|_2 \leq 1} \mathbb{E}y^\top H y,$$

and the property of the $\ell_2$-operator norm,

$$\leq \sup_{\|x\|_2 \leq 1} \mathbb{E}\|H\|_2 \|y\|_2^2 = \sup_{\|x\|_2 \leq 1} \mathbb{E}\|H\|_2 \|Jx\|_2^2.$$

From our assumption about the maximal eigenvalue of $H$,

$$\leq L \sup_{\|x\|_2 \leq 1} \mathbb{E}\|Jx\|_2^2,$$

denoting the $\ell_2$ vector norm as a quadratic form as,

$$= L \sup_{\|x\|_2 \leq 1} x^\top \left(\mathbb{E}J^\top J\right) x,$$

again, by the property of the $\ell_2$-operator norm,

$$\leq L \|\mathbb{E}J^\top J\|_2 \sup_{\|x\|_2 \leq 1} \|x\|_2^2$$

$$= L \|\mathbb{E}J^\top J\|_2.$$

$\qquad\square$

**Lemma 9.** *For a* 1-*Lipschitz diagonal conditioner $\phi$, the Jacobian of the location-scale reparameterization function $\mathcal{T}_\lambda$ satisfies*

$$\left\| \mathbb{E} \left( \frac{\partial \mathcal{T}_\lambda(\boldsymbol{u})}{\partial \lambda} \right)^\top \frac{\partial \mathcal{T}_\lambda(\boldsymbol{u})}{\partial \lambda} \right\|_2 \leq 1.$$

*Proof.* For notational clarity, we will occasionally represent $\mathcal{T}_\lambda$ as

$$\mathcal{T}_\lambda(\boldsymbol{u}) = \mathcal{T}(\lambda; \boldsymbol{u}),$$

such that $\mathcal{T}_i(\lambda; \boldsymbol{u})$ denotes the $i$th component of $\mathcal{T}_\lambda$.

From the definition of $\mathcal{T}_\lambda$, it is straightforward to notice that its Jacobian is the concatenation of 3 block matrices

$$\boldsymbol{J_m} = \frac{\partial \mathcal{T}_\lambda(\boldsymbol{u})}{\partial \boldsymbol{m}}, \qquad \boldsymbol{J_s} = \frac{\partial \mathcal{T}_\lambda(\boldsymbol{u})}{\partial \boldsymbol{s}}, \quad \text{and} \quad \boldsymbol{J_L} = \frac{\partial \mathcal{T}_\lambda(\boldsymbol{u})}{\partial \text{vec}(\boldsymbol{L})}.$$

The $\boldsymbol{m}$ block form a deterministic identity matrix

$$\boldsymbol{J_m} = \frac{\partial \mathcal{T}_\lambda(\boldsymbol{u})}{\partial \boldsymbol{m}} = \boldsymbol{I},$$

which is shown by (Domke, 2020, Lemma 4).

The proof strategy is as follows: we will directly compute the squared Jacobian through block matrix multiplication. The key is that, after expectation, the resulting matrix becomes diagonal. Then, the $\ell_2$ operator norm, or maximal eigenvalue, follows trivially as the maximal diagonal element.

First,

$$
\begin{aligned}
\mathbb{E} \left( \frac{\partial \mathcal{T}(\lambda; \boldsymbol{u})}{\partial \lambda} \right)^\top \frac{\partial \mathcal{T}(\lambda; \boldsymbol{u})}{\partial \lambda} &= \mathbb{E} \begin{bmatrix} \boldsymbol{J_m^\top} \\ \boldsymbol{J_s^\top} \\ \boldsymbol{J_L^\top} \end{bmatrix} \begin{bmatrix} \boldsymbol{J_m} & \boldsymbol{J_s} & \boldsymbol{J_L} \end{bmatrix} \\
&= \mathbb{E} \begin{bmatrix} \boldsymbol{J_m^\top J_m} & \boldsymbol{J_m^\top J_s} & \boldsymbol{J_m^\top J_L} \\ \boldsymbol{J_s^\top J_m} & \boldsymbol{J_s^\top J_s} & \boldsymbol{J_s^\top J_L} \\ \boldsymbol{J_L^\top J_m} & \boldsymbol{J_L^\top J_s} & \boldsymbol{J_L^\top J_L} \end{bmatrix} \\
&= \begin{bmatrix} \boldsymbol{I} & \mathbb{E}\boldsymbol{J_s} & \mathbb{E}\boldsymbol{J_L} \\ \mathbb{E}\boldsymbol{J_s^\top} & \mathbb{E}\boldsymbol{J_s^\top J_s} & \mathbb{E}\boldsymbol{J_s^\top J_L} \\ \mathbb{E}\boldsymbol{J_L^\top} & \mathbb{E}\boldsymbol{J_L^\top J_s} & \mathbb{E}\boldsymbol{J_L^\top J_L} \end{bmatrix}.
\end{aligned}
$$

For $\boldsymbol{J_s}$, the entries are

$$\frac{\partial \mathcal{T}_i(\boldsymbol{u})}{\partial s_j} = \phi'(s_i) u_j \mathbb{1}_{i=j},$$

which is a diagonal matrix. Thus, by Assumption 1,

$$\mathbb{E}\boldsymbol{J_s} = \boldsymbol{O}, \qquad \mathbb{E}\boldsymbol{J_s^\top J_s} = \text{diag}(\boldsymbol{\phi}'(\boldsymbol{s}))^2.$$

For $\boldsymbol{J_s}$, the entries are

$$\frac{\partial \mathcal{T}_i(\lambda; \boldsymbol{u})}{\partial L_{jk}} = u_k \mathbb{1}_{i=j}.$$

To gather some intuition, the case of $d = 4$ looks like the following:

$$\boldsymbol{J_L} = \begin{bmatrix} u_2 & u_3 & u_4 & & & & & & & \\ & & & u_1 & u_3 & u_4 & & & & \\ & & & & & & u_1 & u_2 & u_4 & \\ & & & & & & & & & u_1 & u_2 & u_3 \end{bmatrix}.$$

It is crucial to notice that the $i$th row does *not* include $u_i$. This means that, the matrix $\boldsymbol{J_s^\top J_L}$ has entries that are either 0, or $\phi'(s_i) u_i u_j$ for $i \neq j$, which is $\mathbb{E}\phi'(s_i) u_i u_j = 0$ by Assumption 1. Therefore,

$$\mathbb{E}\boldsymbol{J_s^\top J_L} = \boldsymbol{O}.$$

Finally, the elements of $\boldsymbol{J}_L^\top \boldsymbol{J}_L$ are

$$\mathbb{E} \sum_{i=0}^{d} \frac{\partial \mathcal{T}_i(\lambda; \boldsymbol{u})}{\partial L_{jk}} \frac{\partial \mathcal{T}_i(\lambda; \boldsymbol{u})}{\partial L_{lm}} = \mathbb{E} \sum_{i=0}^{d} u_k u_m \mathbb{1}_{i=j} \mathbb{1}_{i=l} = \mathbb{1}_{j=l} \left(\mathbb{E} u_k u_m\right) = \mathbb{1}_{j=l} \mathbb{1}_{k=m},$$

where the last equality follows from Assumption 1, which forms an identity matrix as

$$\mathbb{E} \boldsymbol{J}_L^\top \boldsymbol{J}_L = \mathbf{I}.$$

Therefore, the expected-squared Jacobian is now

$$\mathbb{E} \left(\frac{\partial \mathcal{T}_\lambda(\boldsymbol{u})}{\partial \lambda}\right)^\top \frac{\partial \mathcal{T}_\lambda(\boldsymbol{u})}{\partial \lambda} = \begin{bmatrix} \mathbf{I} & \mathbb{E}\boldsymbol{J}_s & \mathbb{E}\boldsymbol{J}_L \\ \mathbb{E}\boldsymbol{J}_s^\top & \mathbb{E}\boldsymbol{J}_s^\top \boldsymbol{J}_s & \mathbb{E}\boldsymbol{J}_s^\top \boldsymbol{J}_L \\ \mathbb{E}\boldsymbol{J}_L^\top & \mathbb{E}\boldsymbol{J}_L^\top \boldsymbol{J}_s & \mathbb{E}\boldsymbol{J}_L^\top \boldsymbol{J}_L \end{bmatrix}$$

$$= \begin{bmatrix} \mathbf{I} & & \\ & \mathrm{diag}\left(\boldsymbol{\phi}(\boldsymbol{s})\right)^2 & \\ & & \mathbf{I} \end{bmatrix},$$

which, conveniently, is a diagonal matrix. The maximal singular value of a block-diagonal matrix is the maximal singular value of each block. And since each block is diagonal with only positive entries, the largest element forms the maximal singular value. As we assume that $\phi$ is 1-Lipchitz, the element of all blocks is lower-bounded by 0 and upper-bounded by 1. Therefore, the maximal singular value of the expected-squared Jacobian is bounded by 1. □

**Theorem 1.** *Let $\ell$ be $L_\ell$-smooth and twice differentiable. Then, the following results hold:*

*(i) If $\phi$ is linear, the energy $f$ is $L_\ell$-smooth.*

*(ii) If $\phi$ is 1-Lipschitz, the energy $\ell$ is $(L_\ell + L_s)$-smooth if and only if* Assumption 3 *holds.*

*Proof.* For notational clarity, we will occasionally represent $\mathcal{T}_\lambda$ as

$$\mathcal{T}_\lambda(u) = \mathcal{T}(\lambda; u),$$

such that $\mathcal{T}_i(\lambda; u)$ denotes the $i$th component of $\mathcal{T}_\lambda$.

By the Leibniz and chain rule, the Hessian of the energy $f$ follows as

$$\nabla^2 f(\lambda) = \mathbb{E}\nabla^2_\lambda \ell(\mathcal{T}_\lambda(u))$$

$$= \underbrace{\mathbb{E}\left(\frac{\partial \mathcal{T}_\lambda(u)}{\partial \lambda}\right)^\top \nabla^2 \ell(\mathcal{T}_\lambda(u)) \frac{\partial \mathcal{T}_\lambda(u)}{\partial \lambda}}_{\triangleq T_{\mathrm{lin}}} + \underbrace{\mathbb{E}\sum_{i=1}^d D_i \ell(\mathcal{T}_\lambda(u)) \frac{\partial^2 \mathcal{T}_i(\lambda; u)}{\partial \lambda^2}}_{\triangleq T_{\mathrm{non}}}.$$

When $\mathcal{T}$ is linear with respect to $\lambda$, it is clear that we have

$$\frac{\partial^2 \mathcal{T}_i(\lambda; u)}{\partial \lambda^2} = \mathbf{0}. \tag{5}$$

Then, $T_{\mathrm{non}}$ is zero. In contrast, $T_{\mathrm{lin}}$ appears for both the linear and nonlinear cases. Therefore, $T_{\mathrm{non}}$ fully characterizes the effect of nonlinearity in the reparameterization function.

Now, the triangle inequality yields

$$\|\nabla^2 f(\lambda)\|_2 = \|T_{\mathrm{lin}} + T_{\mathrm{non}}\|_2 \leq \|T_{\mathrm{lin}}\|_2 + \|T_{\mathrm{non}}\|_2,$$

where equality is achieved when either term is 0. On the contrary, the reverse triangle inequality states that

$$\left| \|T_{\mathrm{lin}}\|_2 - \|T_{\mathrm{non}}\|_2 \right| \leq \|\nabla^2 f(\lambda)\|_2.$$

This implies that, if either $T_{\mathrm{lin}}$ or $T_{\mathrm{non}}$ is unbounded, the Hessian is not bounded. Thus, ensuring that $T_{\mathrm{lin}}$ and $T_{\mathrm{non}}$ are bounded is sufficient and necessary to establish that $f$ is smooth.

**Proof of (i)** The bound on the linear part, $T_{\mathrm{lin}}$, follows from Lemma 2 as

$$\|T_{\mathrm{lin}}\|_2 = \left\| \mathbb{E}\left(\frac{\partial \mathcal{T}_\lambda(u)}{\partial \lambda}\right)^\top \nabla^2 \ell(\mathcal{T}_\lambda(u)) \frac{\partial \mathcal{T}_\lambda(u)}{\partial \lambda} \right\|_2$$

$$\leq L_\ell \left\| \mathbb{E}\left(\frac{\partial \mathcal{T}_\lambda(u)}{\partial \lambda}\right)^\top \frac{\partial \mathcal{T}_\lambda(u)}{\partial \lambda} \right\|_2,$$

and from the 1-Lipschitzness of $\phi$, Lemma 9 yields

$$\leq L_\ell.$$

When $\phi$ is linear, it immediately follows from Equation (5) that

$$\|\nabla^2 f(\lambda)\|_2 = \|T_{\mathrm{lin}}\|_2 \leq L_\ell,$$

which is tight as shown by Domke (2020, Theorem 6).

**Proof of (ii)** For the nonlinear part $T_{\mathrm{non}}$, we use the fact that $\mathcal{T}_i(\lambda; u)$ is given as

$$\mathcal{T}_i(\lambda; u) = m_i + \phi(s_i) u_i + \sum_{j \neq i} L_{ij} u_j.$$

The second derivative of $\mathcal{T}_i$ is clearly non-zero only for the nonlinear part involving $s_1, \dots, s_d$. Thus, $T_{\text{non}}$ follows as

$$T_{\text{non}} = \mathbb{E} \sum_{i=1}^{d} \mathrm{D}_i \ell \left( \mathcal{T}_\lambda \left( \boldsymbol{u} \right) \right) \frac{\partial^2 \mathcal{T}_i \left( \lambda; \boldsymbol{u} \right)}{\partial \lambda^2}$$

$$= \mathbb{E} \sum_{i=1}^{d} g_i \left( \lambda; \boldsymbol{u} \right) \frac{\partial^2 \mathcal{T}_i \left( \lambda; \boldsymbol{u} \right)}{\partial \lambda^2}$$

$$= \begin{bmatrix} \cdot & \cdot & \cdot \\ \cdot & \mathbb{E} \sum_{i=1}^{d} g_i \left( \lambda; \boldsymbol{u} \right) \frac{\partial^2 \mathcal{T}_i(\lambda; \boldsymbol{u})}{\partial s^2} & \cdot \\ \cdot & \cdot & \cdot \end{bmatrix}.$$

Furthermore, the second-order derivatives with respect to $s_1, \dots, s_d$ are given as

$$\frac{\partial^2 \mathcal{T}_i \left( \lambda; \boldsymbol{u} \right)}{\partial s_j^2} = \mathbb{1}_{i=j} \phi'' \left( s_j \right).$$

Considering this, the only non-zero block of $T_{\text{non}}$ forms a diagonal matrix as

$$\mathbb{E} \sum_{i=1}^{d} g_i \left( \lambda; \boldsymbol{u} \right) \frac{\partial^2 \mathcal{T}_i \left( \lambda; \boldsymbol{u} \right)}{\partial s} = \begin{bmatrix} \mathbb{E} g_1 \left( \lambda; \boldsymbol{u} \right) \frac{\partial^2 \mathcal{T}_1(\lambda; \boldsymbol{u})}{\partial s_1^2} & & \\ & \ddots & \\ & & \mathbb{E} g_d \left( \lambda; \boldsymbol{u} \right) \frac{\partial^2 \mathcal{T}_d(\lambda; \boldsymbol{u})}{\partial s_d^2} \end{bmatrix}$$

$$= \begin{bmatrix} \mathbb{E} g_1 \left( \lambda; \boldsymbol{u} \right) \phi'' \left( s_1 \right) u_1 & & \\ & \ddots & \\ & & \mathbb{E} g_d \left( \lambda; \boldsymbol{u} \right) \phi'' \left( s_d \right) u_d \end{bmatrix}$$

This implies that the only non-zero entries of $T_{\text{non}}$ lie on its diagonal. Since the $\ell_2$ norm of a diagonal matrix is the value of the maximal diagonal element,

$$\| T_{\text{non}} \|_2 \leq \max_{i=1,\dots,d} \mathbb{E} g_i \left( \lambda; \boldsymbol{u} \right) \phi'' \left( s_i \right) u_i \leq L_s,$$

where $L_s$ is finite constant if Assumption 3 holds. On the contrary, if a finite $L_s$ does not exist, $\| T_{\text{non}} \|_2$ cannot be bounded. Therefore, the energy is smooth if and only if Assumption 3 holds. When it does, the energy $f$ is $L_f + L_s$ smooth. $\qquad\square$

**Example 2.** Let $\ell(z) = (1/2)\, z^\top A z$ and the diagonal conditioner be $\phi(x) = \text{softplus}(x)$. Then,

(i) if $A$ is dense and the variational family is the mean-field family or

(ii) if $A$ is diagonal and the variational family is the Cholesky family,

Assumption 3 holds with $L_s \approx 0.26034\,(\max_{i=1,\dots,d} A_{ii})$.

(iii) If $A$ is dense but the Cholesky family is used, Assumption 3 does not hold.

*Proof.* Since the gradient is

$$\nabla \ell(z) = Az,$$

combined with reparameterization, we have

$$g(\lambda; u) = A(Cu + m)$$

Then, for each coordinate $i = 1, \dots, d$, we have

$$\mathbb{E} g_i(\lambda; u)\, u_i \phi''(s_i) = \mathbb{E}\left( \sum_j \sum_{k \leq j} A_{ij} C_{jk} u_k + \sum_j A_{ij} m_j \right) u_i \phi''(s_i)$$

$$= \sum_j \sum_{k \leq j} A_{ij} C_{jk} \mathbb{E} u_k u_i \phi''(s_i) + \sum_j A_{ij} m_j \mathbb{E} u_i \phi''(s_i),$$

and from Assumption 1,

$$= \phi''(s_i) \sum_j \sum_{k \leq j} A_{ij} C_{jk} \mathbb{1}_{k=i}$$

$$= \phi''(s_i) \sum_j A_{ij} C_{ji}.$$

Furthermore, the diagonal of $C$ involves $\phi$ such that

$$\mathbb{E} g_i(\lambda; u)\, u_i \phi''(s_i) = \underbrace{A_{ii} \phi(s_i)\, \phi''(s_i)}_{T_{\text{diag}}} + \underbrace{\sum_{j<i} A_{ij} C_{ji} \phi''(s_i)}_{T_{\text{off}}}.$$

For the softplus function, we have

$$0 < \phi''(s) < 1$$

for any finite $s$, and we have

$$\sup_s \phi(s)\, \phi''(s) \approx 0.26034,$$

where the supremum was numerically approximated. Then, it is clear that $T_{\text{diag}}$ is finite as long as the diagonals of $A$ are finite. Furthermore, we have the following:

(i) If $A$ is diagonal, then $T_{\text{off}}$ is 0.

(ii) If $A$ is dense but $C$ is diagonal due to the use of the mean-field family, $T_{\text{off}}$ is again 0.

(iii) However, when both $A$ and $C$ are not diagonal, $T_{\text{off}}$ can be made arbitrarily large.

$\square$

### F.2.2 Convexity

**Lemma 10.** *Let $\ell$ be convex. Then, for a convex nonlinear $\phi$, the inequality*

$$\langle \nabla_\lambda f(\lambda), \lambda - \lambda' \rangle \leq \mathbb{E} \langle \nabla g(\lambda; u), \mathcal{T}_\lambda(u) - \mathcal{T}_{\lambda'}(u) \rangle$$

*holds for all $\lambda \in \Lambda$ if and only if [Assumption 4] holds. For the linear parameterization, the inequality becomes equality.*

*Proof.* First, notice that the left-hand side is

$$\langle \nabla_\lambda f(\lambda), \lambda - \lambda' \rangle = \sum_{i=1}^p \langle \nabla_\lambda \mathbb{E}\ell(\mathcal{T}_\lambda(u)), \lambda - \lambda' \rangle = \mathbb{E} \sum_{i=1}^p \left\langle \left( \frac{\partial \mathcal{T}_\lambda(u)}{\partial \lambda_i} \right) g(\lambda; u), \lambda - \lambda' \right\rangle.$$

By restricting us to the location-scale family, we then get

$$= \mathbb{E} \Bigg( \underbrace{\sum_i \left( \frac{\partial \mathcal{T}_\lambda(u)}{\partial m_i} \right) g(\lambda; u)(m_i - m_i')}_{\text{convexity with respect to } m} + \underbrace{\sum_{ij} \left( \frac{\partial \mathcal{T}_\lambda(u)}{\partial L_{ij}} \right) g(\lambda; u)(L_{ij} - L_{ij}')}_{\text{convexity with respect to } L}$$

$$+ \underbrace{\sum_i \mathbb{E}\left( \frac{\partial \mathcal{T}_\lambda(u)}{\partial s_i} \right) g(\lambda; u)(s_i - s_i')}_{\text{convexity with respect to } s} \Bigg),$$

and plugging the derivatives of the reparameterization function,

$$= \mathbb{E} \Bigg( \sum_i g_i(\lambda; u)(m_i - m_i') + \sum_i \sum_{j<i} u_j g_i(\lambda; u)(L_{ij} - L_{ij}') + \sum_i \phi'(s_i) u_i g_i(\lambda; u)(s_i - s_i') \Bigg).$$

On the other hand, the right-hand side follows as

$$\mathbb{E} \langle \nabla g(\lambda; u), \mathcal{T}_\lambda(u) - \mathcal{T}_{\lambda'}(u) \rangle$$

$$= \mathbb{E} \Big( \langle g(\lambda; u), m - m' \rangle + \langle g(\lambda; u), (L - L')u \rangle + \langle g(\lambda; u), (\Phi(s) - \Phi(s'))u \rangle \Big)$$

$$= \mathbb{E} \Bigg( \sum_i g_i(\lambda; u)(m_i - m_i') + \sum_i \sum_{j<i} u_j g_i(\lambda; u)(L_{ij} - L_{ij}') + \sum_i g_i(\lambda; u) u_i(\phi(s_i) - \phi(s_i')) \Bigg).$$

The convexity with respect to the $m$ and $L$ is clear from the first two terms; they are equal. The statement is now up to the last term. That is, the statement holds if

$$\mathbb{E} \sum_i g_i(\lambda; u) u_i \phi'(s_i)(s_i - s_i') \leq \mathbb{E} \sum_i g_i(\lambda; u) u_i(\phi(s_i) - \phi(s_i')). \tag{6}$$

For this, we will show that [Assumption 4] is both necessary and sufficient.

**Proof of sufficiency** [Equation (6)] holds if

$$\mathbb{E} g_i(\lambda; u) u_i \geq 0$$

for all $i = 1, \dots, d$, which is non other than [Assumption 4].

**Proof of necessity** Suppose that the inequality

$$\langle \nabla_\lambda f(\lambda), \lambda - \lambda' \rangle \leq \mathbb{E} \langle g(\lambda; u), \mathcal{T}_\lambda(u) - \mathcal{T}_{\lambda'}(u) \rangle$$

holds for all $\lambda \in \Lambda$, implying

$$\sum_i \mathbb{E} u_i g_i(\lambda; u) \phi'(s_i)(s_i - s_i') \leq \sum_i \mathbb{E} u_i g_i(\lambda; u)((\phi(s_i) - \phi(s_i'))).$$

For any $\lambda$, we are free to set any $\lambda' \in \Lambda$ and check whether we can retrieve [Assumption 4] for this specific $\lambda$. Now, for each axis $i$, set $s_j' = s_j$ for all $j \neq i$, then

$$\mathbb{E} u_i g_i(\lambda; u) \phi'(s_i)(s_i - s_i') \leq \mathbb{E} u_i g_i(\lambda; u)(\phi(s_i) - \phi(s_i')).$$

Since $\phi$ is assumed to be convex such that

$$\phi'(s_i)(s_i - s_i') \le \phi(s_i) - \phi(s_i').$$

it follows that

$$\mathbb{E}\, u_i\, g_i(\lambda; u) \ge 0. \tag{7}$$

Therefore, for any $\lambda \in \Lambda$ it must be that Assumption 4 holds. $\qquad\square$

---

**Proposition 1.** *We have the following:*
  *(i) If $\ell$ is convex, then for the mean-field family, Assumption 4 holds.*
  *(ii) For the Cholesky family, there exists a convex $\ell$ where Assumption 4 does not hold.*

---

*Proof.* For **(i)**, the key property is the monotonicity of the gradient.

**Proof of (i)**   For the mean-field family, recall that

$$C_{ii} = \phi(s_i).$$

Also, observe that

$$Cu + m = (C_{11}u_1 + m_1, \dots, C_{dd}u_d + m_d).$$

By the property of convex functions, $\nabla \ell$ is monotone such that

$$\langle \nabla \ell(z) - \nabla \ell(z'), z - z' \rangle \ge 0.$$

Now, by setting $z = Cu + m$ and $z' = Cu + m - C_{ii}u_i e_i$, we obtain

$$\langle \nabla \ell(Cu + m) - \nabla \ell(Cu + m - C_{ii}u_i e_i), C_{ii}u_i e_i \rangle \ge 0$$

for every $i = 1, \dots, d$.

For the mean-field family, $Cu + m - C_{ii}u_i e_i$ is now independent of $u_i$. Thus,

$$
\begin{aligned}
\mathbb{E}\, C_{ii}u_i \mathrm{D}_i \ell(Cu + m) &\ge \mathbb{E}\, C_{ii}u_i\, \mathrm{D}_i \ell(Cu + m - e_i C_{ii}u_i) \\
&= C_{ii}(\mathbb{E}\, u_i)(\mathbb{E}\, \mathrm{D}_i f(Cu + m - e_i C_{ii}u_i)) \\
&= 0,
\end{aligned}
$$

where $\mathrm{D}_i f$ denotes the $i$th axis of $\nabla f$. Since $C_{ii} > 0$ by design,

$$\mathbb{E}\, C_{ii}u_i \mathrm{D}_i f(Cu + m) > 0 \quad \Leftrightarrow \quad \mathbb{E}\, u_i \mathrm{D}_i f(Cu + m) > 0,$$

which is Assumption 4.

**Proof of (ii)**   We provide an example that proves the statement. Let $\ell(z) = \frac{1}{2}z^\top A z$. Then,

$$g(\lambda; u) = \ell(\mathcal{T}_\lambda(u)) = A(Cu + m) = ACu + Am.$$

Suppose that we choose $\lambda$ such that

$$C = \begin{bmatrix} 1 & 0 \\ 1 & 1 \end{bmatrix}$$

and $m = 0$. Also, setting

$$A = \begin{bmatrix} 1 & -2 \\ -2 & 5 \end{bmatrix},$$

we get a strongly convex function $\ell$. Then,

$$g(\lambda; u) = ACu = \begin{bmatrix} 1 & -2 \\ -2 & 5 \end{bmatrix}\begin{bmatrix} 1 & 0 \\ 1 & 1 \end{bmatrix}\begin{bmatrix} u_1 \\ u_2 \end{bmatrix} = \begin{bmatrix} -1 & -2 \\ 3 & 5 \end{bmatrix}\begin{bmatrix} u_1 \\ u_2 \end{bmatrix} = \begin{bmatrix} -u_1 - 2u_2 \\ 3u_1 + 5u_2 \end{bmatrix}$$

Finally, we have

$$\mathbb{E} g_1(\lambda; u) u_1 = \mathbb{E}(-u_1 - 2u_2) u_1 = -1 < 0,$$

which violates Assumption 4. $\qquad\square$

**Lemma 11.** *For any function $f \in C^1(\mathbb{R}, \mathbb{R}_+)$, there is no constant $0 < L < \infty$ such that*

$$|f(x) - f(y)| \geq L|x - y|.$$

*Proof.* Suppose for the sake of contradiction that such $L > 0$ exists. Letting $y \to x$ gives $|f'(x)| \geq L$ for all $x \in \mathbb{R}$. For each $x$, either $f'(x) \leq -L$ or $f'(x) \geq L$ holds. We discuss two cases based on the value of $f'(0)$.

If $f'(0) \geq L$, we claim that $f'(x) \geq L$ for all $x \in \mathbb{R}$. Otherwise, $f'(x) < L$ for some $x$ implies $f'(x) \leq -L$. By the intermediate value theorem ($f'$ is continuous), there exists a point $y$ between 0 and $x$ that attains the value $f'(y) = 0$, which is a contradiction.

Now that $f'(x) \geq L > 0$ for all $x$, $f$ is an increasing function. For any $x < 0$, we have

$$
\begin{aligned}
f(x) &= f(x) - f(0) + f(0) \\
&= -|f(x) - f(0)| + f(0) \\
&\leq -L|x| + f(0).
\end{aligned}
$$

Here, we can plug $x' = -\frac{f(0)}{L}$ as

$$f(x') = -L\left|-\frac{f(0)}{L}\right| + f(0) = -|f(0)| + f(0) \leq 0,$$

which implies that $f(x') \notin \mathbb{R}_+$, which is a contradiction.

Now we discuss the second case $f'(0) \leq -L$. By a similar argument, $f'(x) \leq -L$ for all $x \in \mathbb{R}$. Thus, $f$ is a decreasing function. For any $x > 0$, we have

$$
\begin{aligned}
f(x) &= f(x) - f(0) + f(0) \\
&= -|f(x) - f(0)| + f(0) \\
&\leq -Lx + f(0).
\end{aligned}
$$

Picking $x' = \frac{f(0)}{L}$ results in $f(x') \notin \mathbb{R}_+$, which is a contradiction. $\square$

**Theorem 2.** *Let $\ell$ be $\mu$-strongly convex. Then, we have the following:*

    *(i) If $\phi$ is linear, the energy $f$ is $\mu$-strongly convex.*

    *(ii) If $\phi$ is convex, the energy $f$ is convex if and only if Assumption 4 holds.*

    *(iii) If $\phi$ is such that $\phi \in \mathrm{C}^1(\mathbb{R}, \mathbb{R}_+)$, the energy $f$ is not strongly convex.*

*Proof.* The special case **(i)** is proven by Domke (2020, Theorem 9). We focus on the general statement **(ii)**.

If $\ell$ is $\mu$-strongly convex, the inequality

$$\ell(\mathbf{z}) - \ell(\mathbf{z}') \geq \langle \nabla \ell(\mathbf{z}'), \mathbf{z} - \mathbf{z}' \rangle + \frac{\mu}{2} \|\mathbf{z} - \mathbf{z}'\|_2^2 \tag{8}$$

holds, where the general convex case is obtained as a special case with $\mu = 0$. The goal is to relate this to the ($\mu$-strong-)convexity of the energy with respect to the variational parameters given by

$$f(\boldsymbol{\lambda}) - f(\boldsymbol{\lambda}') \geq \langle \nabla_{\boldsymbol{\lambda}} f(\boldsymbol{\lambda}'), \boldsymbol{\lambda} - \boldsymbol{\lambda}' \rangle + \frac{\mu}{2} \|\boldsymbol{\lambda} - \boldsymbol{\lambda}'\|_2^2.$$

**Proof of (ii)** Plugging the reparameterized latent variables to Equation (8) and taking the expectation, we have

$$\mathbb{E}\ell(\mathcal{T}_{\boldsymbol{\lambda}}(\mathbf{u})) - \mathbb{E}\ell(\mathcal{T}_{\boldsymbol{\lambda}'}(\mathbf{u})) \geq \mathbb{E}\langle \nabla \ell(\mathcal{T}_{\boldsymbol{\lambda}'}(\mathbf{u})), \mathcal{T}_{\boldsymbol{\lambda}}(\mathbf{u}) - \mathcal{T}_{\boldsymbol{\lambda}'}(\mathbf{u})\rangle + \frac{\mu}{2}\mathbb{E}\|\mathcal{T}_{\boldsymbol{\lambda}}(\mathbf{u}) - \mathcal{T}_{\boldsymbol{\lambda}'}(\mathbf{u})\|_2^2$$

$$\Leftrightarrow \qquad f(\boldsymbol{\lambda}) - f(\boldsymbol{\lambda}') \geq \mathbb{E}\langle \nabla \ell(\mathcal{T}_{\boldsymbol{\lambda}'}(\mathbf{u})), \mathcal{T}_{\boldsymbol{\lambda}}(\mathbf{u}) - \mathcal{T}_{\boldsymbol{\lambda}'}(\mathbf{u})\rangle + \frac{\mu}{2}\mathbb{E}\|\mathcal{T}_{\boldsymbol{\lambda}}(\mathbf{u}) - \mathcal{T}_{\boldsymbol{\lambda}'}(\mathbf{u})\|_2^2$$

$$\Leftrightarrow \qquad f(\boldsymbol{\lambda}) - f(\boldsymbol{\lambda}') \geq \mathbb{E}\langle \nabla g(\boldsymbol{\lambda}'; \mathbf{u}), \mathcal{T}_{\boldsymbol{\lambda}}(\mathbf{u}) - \mathcal{T}_{\boldsymbol{\lambda}'}(\mathbf{u})\rangle + \frac{\mu}{2}\mathbb{E}\|\mathcal{T}_{\boldsymbol{\lambda}}(\mathbf{u}) - \mathcal{T}_{\boldsymbol{\lambda}'}(\mathbf{u})\|_2^2$$

Thus, the energy is convex if and only if

$$\mathbb{E}\langle g(\boldsymbol{\lambda}; \mathbf{u}), \mathcal{T}_{\boldsymbol{\lambda}}(\mathbf{u}) - \mathcal{T}_{\boldsymbol{\lambda}'}(\mathbf{u})\rangle \geq \langle \nabla f(\boldsymbol{\lambda}), \boldsymbol{\lambda} - \boldsymbol{\lambda}'\rangle$$

holds. This is established by Lemma 10.

**Proof of (iii)** We now prove that, under the nonlinear parameterization, the energy cannot be strongly convex. When the energy is convex, it is also strongly convex if and only if

$$\frac{\mu}{2}\mathbb{E}\|\mathcal{T}_{\boldsymbol{\lambda}}(\mathbf{u}) - \mathcal{T}_{\boldsymbol{\lambda}'}(\mathbf{u})\|_2^2 \geq \frac{\mu}{2}\|\boldsymbol{\lambda} - \boldsymbol{\lambda}'\|_2^2.$$

From the proof of Domke (2020, Lemma 5), it follows that

$$\mathbb{E}\|\mathcal{T}_{\boldsymbol{\lambda}}(\mathbf{u}) - \mathcal{T}_{\boldsymbol{\lambda}'}(\mathbf{u})\|_2^2 = \|\mathbf{C} - \mathbf{C}'\|_{\mathrm{F}}^2 + \|\mathbf{m} - \mathbf{m}'\|_2^2.$$

Furthermore, under nonlinear parameterizations,

$$\|\mathbf{C} - \mathbf{C}'\|_{\mathrm{F}}^2 + \|\mathbf{m} - \mathbf{m}'\|_2^2$$

$$= \left\|(\mathbf{D}_\phi(\mathbf{s}) - \mathbf{D}_\phi(\mathbf{s}')) - (\mathbf{L} - \mathbf{L}')\right\|_{\mathrm{F}}^2 + \|\mathbf{m} - \mathbf{m}'\|_2^2,$$

expanding the quadratic,

$$= \left\|\mathbf{D}_\phi(\mathbf{s}) - \mathbf{D}_\phi(\mathbf{s}')\right\|_{\mathrm{F}}^2 + \|\mathbf{L} - \mathbf{L}'\|_{\mathrm{F}}^2 - 2\langle \mathbf{D}_\phi(\mathbf{s}) - \mathbf{D}_\phi(\mathbf{s}'), \mathbf{L} - \mathbf{L}'\rangle_{\mathrm{F}} + \|\mathbf{m} - \mathbf{m}'\|_2^2,$$

and since $\mathbf{D}_\phi(\mathbf{s})$ and $\mathbf{L}$ reside in different sub-spaces, they are orthogonal. Thus,

$$= \left\|\mathbf{D}_\phi(\mathbf{s}) - \mathbf{D}_\phi(\mathbf{s}')\right\|_{\mathrm{F}}^2 + \|\mathbf{L} - \mathbf{L}'\|_{\mathrm{F}}^2 + \|\mathbf{m} - \mathbf{m}'\|_2^2$$

$$= \|\phi(\mathbf{s}) - \phi(\mathbf{s}')\|_2^2 + \|\mathbf{L} - \mathbf{L}'\|_{\mathrm{F}}^2 + \|\mathbf{m} - \mathbf{m}'\|_2^2. \tag{9}$$

For the energy term to be strongly convex, Equation (9) must be bounded *below* by $\|\boldsymbol{\lambda} - \boldsymbol{\lambda}'\|_2^2$. Evidently, this implies that a necessary and sufficient condition is that

$$|\phi(s_{ii}) - \phi(s'_{ii})| \geq L |s_{ii} - s'_{ii}|$$

by some constant $0 < L < \infty$. Notice that the direction of the inequality is reversed from the Lipschitz condition. Unfortunately, there is no such continuous and differentiable function $\phi : \mathbb{R} \to \mathbb{R}_+$, as established by Lemma 11. Thus, for any diagonal conditioner $\phi \in \mathrm{C}^1(\mathbb{R}, \mathbb{R}_+)$, the energy cannot be strongly convex. $\qquad\square$

### F.3 Convergence of Black-Box Variational Inference

#### F.3.1 Vanilla Black-Box Variational Inference

**Theorem 5.** *Let the variational family satisfy Assumption 2, the likelihood satisfy Assumption 5, and the assumptions of Corollary 1 hold such that the ELBO, $F$, is $L_F$-smooth with $L_F = L_\ell + L_\phi + L_s$. Then, if the stepsize satisfy $\gamma < 1/L_F$, the iterates of BBVI with SGD and the M-sample reparameterization gradient estimator satisfy*

$$\min_{0 \leq t \leq T-1} \mathbb{E}\|\nabla F(\lambda_t)\|_2^2 \leq \gamma \frac{2L_F L_\ell \kappa \, C(d,\varphi)}{M} \left( \|\bar{z}_{\text{joint}} - \bar{z}_{\text{like}}\|_2^2 + 2(F^* - f_L^*) \right)$$

$$+ \frac{2}{\gamma T} \left( 1 + \gamma^2 \frac{4L_F L_\ell \, \kappa}{M} C(d,\varphi) \right)^T (F(\lambda_0) - F^*).$$

*where*

| | | |
|---|---|---|
| $\bar{z}_{\text{joint}} = \text{proj}_\ell(z)$ | *is the projection of $z$ onto set of minimizers of $\ell$* | |
| $\bar{z}_{\text{like}} = \text{proj}_{\ell_{\text{like}}}(z)$ | *is the projection of $z$ onto set of minimizers of $\ell_{\text{like}}$,* | |
| $\kappa = L_\ell/\mu$ | *is the condition number,* | |
| $F^* = \inf_{\lambda \in \Lambda} F(\lambda),$ | | |
| $\ell_{\text{like}}^* = \inf_{\lambda \in \mathbb{R}^d} \ell_{\text{like}}(z),$ | | |
| $C(d,\varphi) = d + k_\varphi$ | *for the Cholesky nonlinear,* | |
| $C(d,\varphi) = 2k_\varphi\sqrt{d} + 1$ | *for the mean-field nonlinear,* | |
| $M$ | *is the number of Monte Carlo samples.* | |

*Proof.* Khaled & Richtárik (2023, Theorem 2) show that, if the objective function $F$ is $L_F$-smooth and the stochastic gradients satisfy the *ABC* given as

$$\mathbb{E}\|\widehat{\nabla F}(\lambda)\|_2^2 \leq A(F(\lambda) - F^*) + B\|\nabla F\|_2^2 + C$$

for some $0 < A, B, C < \infty$, SGD guarantees

$$\min_{0 \leq t \leq T-1} \mathbb{E}\|\nabla F(\lambda_t)\|_2^2 \leq L_F C\gamma + \frac{2(1 + L_F \gamma^2 A)^T}{\gamma T}(F(\lambda_0) - F^*).$$

Under the conditions of Corollary 1, $F$ is $L_F$-smooth with $L_F = L_\ell + L_s + L_\phi$. Furthermore, under Assumption 5, Kim *et al.* (2023) show that the Monte Carlo gradient estimates satisfy

$$\mathbb{E}\|\widehat{\nabla F}(\lambda)\|_2^2 \leq \frac{4L_\ell^2 C(d,\varphi)}{\mu M}(F(\lambda) - F^*) + B\|\nabla F\|_2^2$$

$$+ \frac{2L_\ell^2 C(d,\varphi)}{\mu M}\|\bar{z}_{\text{joint}} - \bar{z}_{\text{like}}\|_2^2 + \frac{4L_\ell^2 C(d,\varphi)}{\mu M}(F^* - \ell_{\text{like}}^*),$$

This means that the *ABC* condition is satisfied with constants

$$A = \frac{4L_\ell^2}{\mu M}C(d,\varphi), \qquad B = 1, \qquad C = \frac{2L_\ell^2}{\mu M}C(d,\varphi)\|\bar{z}_{\text{joint}} - \bar{z}_{\text{like}}\|_2^2 + \frac{4L_\ell^2}{\mu M}C(d,\varphi)(F^* - \ell_{\text{like}}^*).$$

Plugging these constants in, we obtain

$$\min_{0 \leq t \leq T-1} \mathbb{E}\|\nabla f(\lambda_t)\|_2^2 \leq \gamma \frac{2L_F L_\ell^2 C(d,\varphi)}{\mu M} \left( \|\bar{z}_{\text{joint}} - \bar{z}_{\text{like}}\|_2^2 + 2(F^* - \ell_{\text{like}}^*) \right)$$

$$+ \frac{2}{\gamma T} \left( 1 + \gamma^2 L_F \frac{4L_\ell^2}{\mu M}C(d,\varphi) \right)^T (F(\lambda_0) - F^*).$$

Substituting the condition number yields the stated result. $\qquad\square$

**Theorem 3.** *Let Assumption 2 hold, the likelihood satisfy Assumption 5, and the assumptions of Corollary 1 hold such that the ELBO F is $L_F$-smooth with $L_F = L_\ell + L_\phi + L_s$. Then, the iterates generated by BBVI through Equation (1) and the M-sample reparameterization gradient include an $\epsilon$-stationary point such that $\min_{0 \le t \le T-1} \mathbb{E}\|\nabla F(\lambda_t)\|_2 \le \epsilon$ for any $\epsilon > 0$ if*

$$T \ge \mathcal{O}\left(\frac{(F(\lambda_0) - F^*)^2 L_F L_\ell^2 C(d, k_\varphi)}{\mu M \epsilon^4}\right)$$

*for some fixed stepsize $\gamma$, where $C(d, \varphi) = d + k_\varphi$ for the Cholesky family and $C(d, \varphi) = 2k_\varphi\sqrt{d} + 1$ for the mean-field family.*

*Proof.* As a corollary to Theorem 5, Khaled & Richtárik (2023, Corollary 1) show that, for an $L_F$-smooth objective function $F$, a gradient estimator satisfying the *ABC* condition, an $\epsilon$-stationary point can be encountered if

$$\gamma = \min\left(\frac{1}{\sqrt{L_F A T}}, \frac{1}{L_F B}, \frac{\epsilon}{2L_F C}\right), \qquad T \ge \frac{12(F(\lambda_0) - F^*)L_F}{\epsilon^2}\max\left(B, \frac{12(F(\lambda_0) - F^*)A}{\epsilon^2}, \frac{2C}{\epsilon^2}\right).$$

Under Assumption 5, Kim *et al.* (2023) show that the Monte Carlo gradient estimates satisfy

$$\mathbb{E}\|\widehat{\nabla}F(\lambda)\|_2^2 \le \frac{4L_\ell^2 C(d, \varphi)}{\mu M}(F(\lambda) - F^*) + B\|\nabla F\|_2^2$$
$$+ \frac{2L_\ell^2 C(d, \varphi)}{\mu M}\|\bar{z}_{\text{joint}} - \bar{z}_{\text{like}}\|_2^2 + \frac{4L_\ell^2 C(d, \varphi)}{\mu M}(F^* - \ell_{\text{like}}^*),$$

This means that the *ABC* condition is satisfied with constants

$$A = \frac{4L_f^2}{\mu M}C(d, \varphi), \qquad B = 1, \qquad C = \frac{2L_f^2}{\mu M}C(d, \varphi)\left(\|\bar{z}_{\text{joint}} - \bar{z}_{\text{like}}\|_2^2 + 2(F^* - f_L^*)\right).$$

where

$$\bar{z}_{\text{joint}} = \text{proj}_\ell(z) \qquad \text{is the projection of } z \text{ onto set of minimizers of } \ell$$
$$\bar{z}_{\text{like}} = \text{proj}_{\ell_{\text{like}}}(z) \qquad \text{is the projection of } z \text{ onto set of minimizers of } \ell_{\text{like}},$$
$$F^* = \inf_{\lambda \in \Lambda} F(\lambda),$$
$$\ell_{\text{like}}^* = \inf_{\lambda \in \mathbb{R}^d} \ell_{\text{like}}(z),$$
$$C(d, \varphi) = d + k_\varphi \qquad \text{for the Cholesky family,}$$
$$C(d, \varphi) = 2k_\varphi\sqrt{d} + 1 \qquad \text{for the mean-field family,}$$
$$M \qquad \text{is the number of Monte Carlo samples.}$$

Plugging these constants in, we obtain

$$T \ge \frac{12(F(\lambda_0) - F^*)L_F}{\epsilon^2}\max\left(1, \frac{48(F(\lambda_0) - F^*)L_\ell^2 C(d, \varphi)}{\mu M \epsilon^2}, \frac{8L_\ell^2 C(d, \varphi)\left(\|\bar{z}_{\text{joint}} - \bar{z}_{\text{like}}\|_2^2 + (F^* - \ell_{\text{like}}^*)\right)}{\mu M \epsilon^2}\right)$$

$$= \mathcal{O}\left(\frac{(F(\lambda_0) - F^*)^2 L_F L_\ell^2 C(d)}{\mu M \epsilon^4}\right),$$

where we omitted the dependence on $k_\varphi$ and the minimizers of $\ell$ and $\ell_{\text{like}}$. $\qquad \square$

### F.3.2 Proximal Black-Box Variational Inference

**Lemma 3** (**Convex Expected Smoothness**). *Let $\ell$ be $L_\ell$-smooth and $\mu$-strongly convex with the variational family satisfying Assumption 2 with the linear parameterization. Then,*

$$\mathbb{E}\|\nabla_\lambda f(\lambda; \boldsymbol{u}) - \nabla_{\lambda'} f(\lambda'; \boldsymbol{u})\|_2^2 \leq 2L_\ell \kappa\, C(d, \varphi)\, \mathrm{B}_f(\lambda, \lambda')$$

*holds, where $\mathrm{B}_f(\lambda, \lambda') \triangleq f(\lambda) - f(\lambda') - \langle \nabla f(\lambda'), \lambda - \lambda' \rangle$ is the Bregman divergence, $\kappa = L_\ell/\mu$ is the condition number, $C(d, \varphi) = d + k_\varphi$ for the Cholesky family, and $C(d, \varphi) = 2k_\varphi\sqrt{d} + 1$ for the mean-field family.*

*Proof.* First, we have

$$\mathbb{E}\|\nabla_\lambda f(\lambda; \boldsymbol{u}) - \nabla_{\lambda'} f(\lambda'; \boldsymbol{u})\|_2^2 = \mathbb{E}\|\nabla_\lambda \ell(\mathcal{T}_\lambda(\boldsymbol{u})) - \nabla_{\lambda'} \ell(\mathcal{T}_{\lambda'}(\boldsymbol{u}))\|_2^2$$

$$= \mathbb{E}\left\| \frac{\partial \mathcal{T}_\lambda(\boldsymbol{u})}{\partial \lambda} g(\lambda, \boldsymbol{u}) - \frac{\partial \mathcal{T}_{\lambda'}(\boldsymbol{u})}{\partial \lambda'} g(\lambda', \boldsymbol{u}) \right\|_2^2.$$

For the linear parameterization, the Jacobian of $\mathcal{T}_\lambda$ does not depend on $\lambda$. Therefore,

$$= \mathbb{E}\left\| \frac{\partial \mathcal{T}_\lambda(\boldsymbol{u})}{\partial \lambda} (g(\lambda, \boldsymbol{u}) - g(\lambda', \boldsymbol{u})) \right\|_2^2$$

and Lemma 6 yields

$$= J_{\mathcal{T}}(\boldsymbol{u})\, \mathbb{E}\|g(\lambda, \boldsymbol{u}) - g(\lambda', \boldsymbol{u})\|_2^2,$$

where

$$J_{\mathcal{T}}(\boldsymbol{u}) = 1 + \|\boldsymbol{u}\|_2^2 \qquad \text{for the Cholesky family and}$$
$$J_{\mathcal{T}}(\boldsymbol{u}) = 1 + \|\boldsymbol{U}^2\|_{\mathrm{F}} \qquad \text{for the mean-field family.}$$

From now on, we apply the strategy of Domke (2019, Theorem 3) for resolving the randomness $\boldsymbol{u}$. That is,

$$\mathbb{E}J_{\mathcal{T}}(\boldsymbol{u})\|g(\lambda, \boldsymbol{u}) - g(\lambda', \boldsymbol{u})\|_2^2 = J_{\mathcal{T}}(\boldsymbol{u})\|\nabla \ell(\mathcal{T}_\lambda(\boldsymbol{u})) - \nabla \ell(\mathcal{T}_{\lambda'}(\boldsymbol{u}))\|_2^2$$

from the $L_\ell$-smoothness of $f$,

$$\leq L_\ell^2\, \mathbb{E}J_{\mathcal{T}}(\boldsymbol{u})\|\mathcal{T}_\lambda(\boldsymbol{u}) - \mathcal{T}_{\lambda'}(\boldsymbol{u})\|_2^2,$$

and applying Corollary 2,

$$\leq L_\ell^2\, C(d, \varphi)\, \|\lambda - \lambda'\|_2^2$$

The last step follows the approach of Kim *et al.* (2023), where we convert the quadratic bound into a bound involving the energy. Recall that the $\mu$-strongly convexity of $\ell$ implies

$$\frac{\mu}{2}\|\boldsymbol{z}' - \boldsymbol{z}\|_2^2 \leq \ell(\boldsymbol{z}) - \ell(\boldsymbol{z}') - \langle \nabla \ell(\boldsymbol{z}'), \boldsymbol{z} - \boldsymbol{z}' \rangle. \tag{10}$$

From Lemma 8, we have

$$L_\ell^2\, C(d, \varphi)\, \|\lambda - \lambda'\|_2^2 = L_f^2\, C(d, \varphi)\, \mathbb{E}\|\mathcal{T}_\lambda(\boldsymbol{u}) - \mathcal{T}_{\lambda'}(\boldsymbol{u})\|_2^2,$$

and by $\mu$-strongly convexity,

$$\leq \frac{2L_\ell^2}{\mu}\, C(d, \varphi)\, \mathbb{E}\big(\ell(\mathcal{T}_\lambda(\boldsymbol{u})) - \ell(\mathcal{T}_{\lambda'}(\boldsymbol{u})) - \langle \nabla \ell(\mathcal{T}_{\lambda'}(\boldsymbol{u})), \mathcal{T}_\lambda(\boldsymbol{u}) - \mathcal{T}_{\lambda'}(\boldsymbol{u}) \rangle\big)$$

$$= \frac{2L_\ell^2}{\mu}\, C(d, \varphi)\, \mathbb{E}\big(f(\lambda; \boldsymbol{u}) - f(\lambda'; \boldsymbol{u}) - \langle g(\lambda'; \boldsymbol{u}), \mathcal{T}_\lambda(\boldsymbol{u}) - \mathcal{T}_{\lambda'}(\boldsymbol{u}) \rangle\big)$$

$$= \frac{2L_\ell^2}{\mu}\, C(d, \varphi)\, \big(f(\lambda) - f(\lambda') - \mathbb{E}\langle g(\lambda'; \boldsymbol{u}), \mathcal{T}_\lambda(\boldsymbol{u}) - \mathcal{T}_{\lambda'}(\boldsymbol{u}) \rangle\big).$$

Finally, by applying the equality in Lemma 10,

$$= \frac{2L_\ell^2}{\mu}\, C(d, \varphi)\, \big(f(\lambda) - f(\lambda') - \langle \nabla f(\lambda'), \lambda - \lambda' \rangle\big).$$

$\square$

**Lemma 12 (Variance Transfer).** *Let $\ell$ be $L_\ell$-smooth and $\mu$-strongly convex with the variational family satisfying Assumption 2 with the linear parameterization. Also, let $\widehat{\nabla}f$ be an M-sample gradient estimator of the energy. Then,*

$$\mathrm{tr}\,\mathbb{V}\,\widehat{\nabla}f(\lambda) \le \frac{4L_\ell \kappa\, C(d,\varphi)}{M}\,\mathrm{B}_f(\lambda,\lambda') + 2\,\mathrm{tr}\,\mathbb{V}\,\widehat{\nabla}f(\lambda'),$$

*$\kappa = L_\ell/\mu$ is the condition number, $\mathrm{B}_f$ is the Bregman divergence defined in Lemma 3, $C(d,\varphi) = d + k_\varphi$ for the Cholesky family, and $C(d,\varphi) = 2k_\varphi\sqrt{d}+1$ for the mean-field family.*

*Proof.* First, the $M$-sample gradient estimator is defined as

$$\widehat{\nabla}f(\lambda) = \frac{1}{M}\sum_{m=1}^{M}\nabla_\lambda f(\lambda; \boldsymbol{u}_m),$$

where $\boldsymbol{u}_m \sim \varphi$. Since $\boldsymbol{u}_1, \dots, \boldsymbol{u}_m$ are independent and identically distributed, we have

$$\mathrm{tr}\,\mathbb{V}\,\widehat{\nabla}f(\lambda) = \frac{1}{M}\mathrm{tr}\,\mathbb{V}\,\nabla_\lambda f(\lambda; \boldsymbol{u}).$$

From here, given Lemma 3, the proof is identical with that of Garrigos & Gower (2023, Lemma 8.20), except for the constants. ◻

**Theorem 6.** *Let $\ell$ be $L_\ell$-smooth and $\mu$-strongly convex. Then, BBVI with proximal SGD in Equation (2), M-Monte Carlo samples, a variational family satisfying Assumption 2, the linear parameterization, and a fixed stepsize $0 < \gamma \le \frac{M}{2L_\ell \kappa C(d,\varphi)}$, the iterates satisfy*

$$\mathbb{E}\|\lambda_T - \lambda^*\|_2^2 \le (1-\gamma\mu)^T\|\lambda_0 - \lambda^2\|_2^2 + \frac{2\gamma\sigma^2}{\mu},$$

*where $\kappa = L_\ell/\mu$ is the condition number, $\sigma^2$ is defined in Lemma 4, $\lambda^* = \arg\min_{\lambda\in\Lambda}F(\lambda)$, $C(d,\varphi) = d + k_\varphi$ for the Cholesky family, and $C(d,\varphi) = 2k_\varphi\sqrt{d}+1$ for the mean-field family.*

*Proof.* Provided that

**(A.6.1)** the energy $f$ is $\mu$-strongly convex,
**(A.6.2)** the energy $f$ is $L_\ell$-smooth,
**(A.6.3)** the regularizer $h$ is convex,
**(A.6.4)** the regularizer $h$ is lower semi-continuous,
**(A.6.5)** the convex expected smoothness condition holds,
**(A.6.6)** the variance transfer condition holds, and
**(A.6.7)** the gradient variance $\sigma^2$ at the optimum is finite such that $\sigma^2 < \infty$,

the proof is identical to that of Garrigos & Gower (2023, Theorem 11.9), which is based on the results of Gorbunov *et al.* (2020, Corollary A.2).

In our setting,

**(A.6.1)** is established by Theorem 2,
**(A.6.2)** is established by Theorem 1,
**(A.6.3)** is trivially satisfied since $h$ is the negative entropy,
**(A.6.4)** is trivially satisfied since $h$ is continuous,
**(A.6.5)** is established in Lemma 3,
**(A.6.6)** is established in Lemma 12,
**(A.6.7)** is established in Lemma 4.

The only difference is that, we replace the constant $L_{\max}$ in the proof of Garrigos & Gower to $L_\ell \kappa C(d,\varphi)/M$. This stems from the different constants in the variance transfer condition. ◻

**Theorem 7.** *Let $\ell$ be $L_\ell$-smooth and $\mu$-strongly convex. Then, for any $\epsilon > 0$, BBVI with proximal SGD in Equation (2), M-Monte Carlo samples, a variational family satisfying Assumption 2, and the linear parameterization guarantees $\mathbb{E}\|\lambda_T - \lambda^*\|_2^2 \leq \epsilon$ if*

$$\gamma = \min\left(\frac{\epsilon}{2}\frac{\mu}{2\sigma^2}, \frac{M}{2L_\ell \kappa C(d,\varphi)}\right), \qquad T \geq \max\left(\frac{1}{\epsilon}\frac{4\sigma^2}{\mu^2}, \frac{2\kappa^2 C(d,\varphi)}{M}\right)\log\left(\frac{2\|\lambda_0 - \lambda^*\|}{\epsilon}\right),$$

*where $\kappa = L_\ell/\mu$, $\sigma^2$ is defined in Lemma 4, $\lambda^* = \arg\min_{\lambda \in \Lambda} F(\lambda)$, $C(d,\varphi) = d + k_\varphi$ for the Cholesky family, and $C(d,\varphi) = 2k_\varphi\sqrt{d} + 1$ for the mean-field family.*

*Proof.* This is a corollary of the fixed stepsize convergence guarantee in Theorem 6 as shown by Garrigos & Gower (2023, Corollary 11.10). They guarantee an $\epsilon$-accurate solution as long as

$$\gamma = \min\left(\frac{\epsilon}{2}\frac{2}{2\sigma_F^*}, \frac{1}{2L_{\max}}\right), \quad T \geq \max\left(\frac{1}{\epsilon}\frac{4\sigma_F^*}{\mu^2}, \frac{2L_{\max}}{\mu}\right)\log\left(\frac{2\|\lambda_0 - \lambda^*\|}{\epsilon}\right).$$

In our notation, $\sigma_F^* = \sigma^2$ and $L_{\max} = L_\ell \kappa C(d,\varphi)/M$. □

**Theorem 8.** *Let $\ell$ be $L_\ell$-smooth and $\mu$-strongly convex. Then, BBVI with proximal SGD in Equation (2), the M-sample reparameterization gradient estimator, a variational family satisfying Assumption 2, the linear parameterization, $T \geq 4T_\kappa$, and a stepsize schedule of*

$$\gamma_t = \begin{cases} \dfrac{M}{2L_\ell \kappa C(d,\varphi)} & \text{for} \quad t \leq 4T_\kappa \\ \dfrac{2t+1}{(t+1)^2\mu} & \text{for} \quad t > 4T_\kappa, \end{cases}$$

*where $T_\kappa = \lceil \kappa^2 C(d,\varphi) M^{-1} \rceil$, $\kappa = L_\ell/\mu$ is the condition number, $C(d,\varphi) = d + k_\varphi$ for the Cholesky family, and $C(d,\varphi) = 2k_\varphi\sqrt{d} + 1$ for the mean-field family, then the iterates satisfy*

$$\mathbb{E}\|\lambda_T - \lambda^*\|_2^2 \leq \frac{16\,T_\kappa^2\,\|\lambda_0 - \lambda^*\|_2^2}{\mathrm{e}^2 T^2} + \frac{8\sigma^2}{\mu^2 T}$$

*where $\sigma^2$ is defined in Lemma 4, $\mathrm{e}$ is Euler's constant, and $\lambda^* = \arg\min_{\lambda \in \Lambda} F(\lambda)$.*

*Proof.* Under our assumptions, Theorem 6 holds, of which the proof is essentially obtaining the recursion

$$\mathbb{E}\|\lambda_{t+1} - \lambda^*\|_2^2 = (1 - \gamma_t\mu)\,\mathbb{E}\|\lambda_t - \lambda^*\|_2^2 + 2\gamma_t^2\sigma^2.$$

Instead of a fixed stepsize, we can apply the decreasing stepsize rule in the proof statement, then which the proof becomes identical to that of Gower et al. (2019, Theorem 3.2). We only need to replace $\mathcal{L}$ with $L_{\max}$ in the proof of Garrigos & Gower (2023, Theorem 11.9). This, in our notation, is $L_{\max} = L_\ell \kappa C(d,\varphi)/M$. □

**Theorem 4.** *Let $\ell$ be $L_\ell$-smooth and $\mu$-strongly convex. Then, for any $\epsilon > 0$, BBVI with proximal SGD in Equation (2), the M-sample reparameterization gradient estimator, a variational family satisfying Assumption 2 with the linear parameterization guarantees $\mathbb{E}\|\lambda_T - \lambda^*\|_2^2 \leq \epsilon$ if*

$$\gamma_t = \begin{cases} \dfrac{M}{2L_\ell \kappa C(d,\varphi)} & \text{for} \quad t \leq 4T_\kappa \\ \dfrac{2t+1}{(t+1)^2\mu} & \text{for} \quad t > 4T_\kappa, \end{cases} \qquad T \geq \max\left(\frac{8\sigma^2}{\mu^2\epsilon} + \frac{4T_\kappa\|\lambda_0 - \lambda^*\|_2}{\mathrm{e}\sqrt{\epsilon}},\ 4T_\kappa\right)$$

*where $\sigma^2$ is defined in Lemma 4, $T_\kappa = \lceil \kappa^2 C(d,\varphi) M^{-1} \rceil$, $\kappa = L_\ell/\mu$ is the condition number, $\mathrm{e}$ is Euler's constant, $\lambda^* = \arg\min_{\lambda \in \Lambda} F(\lambda)$, $C(d,\varphi) = d + k_\varphi$ for the Cholesky family, and $C(d,\varphi) = 2k_\varphi\sqrt{d} + 1$ for the mean-field family.*

*Proof.* The computational complexity follows from the smallest number of iterations $T$ such that

$$\mathbb{E}\|\lambda_T - \lambda^*\|_2^2 \leq \frac{16T_\kappa^2\|\lambda_0 - \lambda^*\|_2^2}{\mathrm{e}^2 T^2} + \frac{8\sigma^2}{\mu^2 T} \leq \epsilon$$

By multiplying both sides with $T^2$ as

$$T^2\epsilon - \frac{8\sigma^2}{\mu^2}T - \frac{16T_\kappa^2\|\lambda_0 - \lambda^*\|_2^2}{e^2} \geq 0, \tag{11}$$

we can see that we are looking for the smallest positive integer that is larger than the solution of a quadratic equation with respect to $T$. This is given as

$$T \geq \frac{\frac{8\sigma^2}{\mu^2} + \sqrt{\left(\frac{8\sigma^2}{\mu^2}\right)^2 + 64\epsilon\frac{T_\kappa^2\|\lambda_0 - \lambda^*\|_2^2}{e^2}}}{2\epsilon}.$$

Applying the inequality $\sqrt{a + b} \leq \sqrt{a} + \sqrt{b}$,

$$\frac{\frac{8\sigma^2}{\mu^2} + \sqrt{\left(\frac{8\sigma^2}{\mu^2}\right)^2 + 64\epsilon\frac{T_\kappa^2\|\lambda_0 - \lambda^*\|_2^2}{e^2}}}{2\epsilon} \leq \frac{\frac{8\sigma^2}{\mu^2} + \left(\frac{8\sigma^2}{\mu^2}\right) + \sqrt{64\epsilon\frac{T_\kappa^2\|\lambda_0 - \lambda^*\|_2^2}{e^2}}}{2\epsilon}$$

$$= \frac{\frac{16\sigma^2}{\mu^2} + \sqrt{\epsilon}\frac{8T_\kappa\|\lambda_0 - \lambda^*\|_2}{e}}{2\epsilon}$$

$$= \frac{8\sigma^2}{\mu^2\epsilon} + \frac{4T_\kappa\|\lambda_0 - \lambda^*\|_2}{e\sqrt{\epsilon}}.$$

Thus, $\mathbb{E}\|\lambda_T - \lambda^*\|_2^2 \leq \epsilon$ can be satisfied with a number of iterations at least

$$T \geq \max\left(\frac{8\sigma^2}{\mu^2\epsilon} + \frac{4T_\kappa\|\lambda_0 - \lambda^*\|_2}{e\sqrt{\epsilon}}, \ 4T_\kappa\right).$$

$\square$

# G Details of Experimental Setup

Table 2: Summary of Datasets and Problems

| Abbrev. | Model | Dataset | $d$ | $N$ |
|---|---|---|---|---|
| LME-election | Linear Mixed Effects | 1988 U.S. presidential election (Gelman & Hill, 2007) | 90 | 11,566 |
| LME-radon | | U.S. household radon levels (Gelman & Hill, 2007) | 391 | 12,573 |
| BT-tennis | Bradley-Terry | ATP World Tour tennis | 6030 | 172,199 |
| LR-keggu | | KEGG-undirected (Shannon et al., 2003) | 31 | 63,608 |
| LR-song | Linear Regression | million songs (Bertin-Mahieux et al., 2011) | 94 | 515,345 |
| LR-buzz | | buzz in social media (Kawala et al., 2013) | 81 | 583,250 |
| LR-electric | | household electric | 15 | 2,049,280 |
| AR-ecg | Sparse Autoregression | Long-term ST ECG (Jager et al., 2003) | 63 | 20,642,000 |

**Linear Regression (LR-\*)**  We consider a basic Bayesian hierarchical linear regression model

$$\sigma_\alpha \sim \mathcal{N}_+\left(0, 10^2\right), \quad \sigma_\beta \sim \mathcal{N}_+\left(0, 10^2\right), \quad \sigma \sim \mathcal{N}_+\left(0, 0.3^2\right)$$
$$\boldsymbol{\beta} \sim \mathcal{N}\left(\mathbf{0}, \sigma_\beta^2 \boldsymbol{I}\right), \quad \alpha \sim \mathcal{N}\left(0, \sigma_\alpha^2\right),$$
$$y_i \sim \mathcal{N}\left(\boldsymbol{\beta}^\top \boldsymbol{x}_i + \alpha, \sigma^2\right),$$

where a weakly informative half-normal hyperprior $\mathcal{N}_+$, a normal distribution with the support restricted to $\mathbb{R}_+$, is assigned on the hyperparameters. For the datasets, we consider large-scale regression problems obtained from the UCI repository (Dua & Graff, 2017), shown in Table 2. For all datasets, we standardize the regressors $\boldsymbol{x}_i$ and the outcomes $y_i$.

**Radon Levels (MLE-radon)**  MLE-radon is a radon level regression problem by Gelman & Hill (2007). It fits a hierarchical mixed-effects model for estimating household radon levels across different counties while considering the floor elevation of each site. The model is described as

$$\sigma \sim \mathcal{N}_+\left(0, 1^2\right), \quad \sigma_\alpha \sim \mathcal{N}_+\left(0, 1^2\right), \quad \mu_\alpha \sim \mathcal{N}\left(0, 10^2\right), \quad \boldsymbol{\epsilon} \sim \mathcal{N}\left(\mathbf{0}, 10^2 \boldsymbol{I}\right)$$
$$\beta_1 \sim \mathcal{N}\left(0, 10^2\right), \quad \beta_2 \sim \mathcal{N}\left(0, 10^2\right)$$
$$\boldsymbol{\alpha} = \mu_\alpha + \sigma_\alpha \boldsymbol{\epsilon}$$
$$\mu_i = \alpha[\text{county}_i] + \beta_1 \log\left(\text{uppm}_i\right) + \text{floor}_i \beta_2$$
$$\log \text{radon}_i \sim \mathcal{N}\left(\mu_i, \sigma^2\right),$$

which uses variable slopes and intercepts with non-centered parameterization. The dataset was obtained from PosteriorDB (Magnusson et al., 2022). Also, for the radon regression problem, the Minnesota subset is often used due to computational reasons. Here, we use the full national dataset.

**Presidential Election (MLE-election)**  MLE-election is a model for studying the effects of sociological factors on the 1988 United States presidential election (Gelman & Hill, 2007). The model is described as

$$\sigma_{\text{age}} \sim \mathcal{N}\left(0, 100^2\right), \quad \sigma_{\text{edu}} \sim \mathcal{N}\left(0, 100^2\right), \quad \sigma_{\text{age}\times\text{edu}} \sim \mathcal{N}\left(0, 100^2\right)$$
$$\sigma_{\text{state}} \sim \mathcal{N}\left(0, 100^2\right), \quad \sigma_{\text{region}} \sim \mathcal{N}\left(0, 100^2\right),$$

$$\boldsymbol{b}_{\text{age}} \sim \mathcal{N}\left(\mathbf{0}, \sigma_{\text{age}}^2 \boldsymbol{I}\right), \quad \boldsymbol{b}_{\text{edu}} \sim \mathcal{N}\left(\mathbf{0}, \sigma_{\text{edu}}^2 \boldsymbol{I}\right), \quad \boldsymbol{b}_{\text{age}\times\text{edu}} \sim \mathcal{N}\left(\mathbf{0}, \sigma_{\text{age}\times\text{edu}}^2 \boldsymbol{I}\right),$$
$$\boldsymbol{b}_{\text{state}} \sim \mathcal{N}\left(\mathbf{0}, \sigma_{\text{state}}^2 \boldsymbol{I}\right), \quad \boldsymbol{b}_{\text{region}} \sim \mathcal{N}\left(\mathbf{0}, \sigma_{\text{region}}^2 \boldsymbol{I}\right)$$

$$\boldsymbol{\beta} \sim \mathcal{N}\left(\mathbf{0}, 100^2 \boldsymbol{I}\right)$$
$$p_i = \beta_1 + \beta_2 \, \text{black}_i + \beta_3 \, \text{female}_i + \beta_4 \, v_{\text{prev},i} + \beta_5 \, \text{female}_i \, \text{black}_i$$
$$+ b_{\text{age}}[\text{age}_i] + b_{\text{edu}}[\text{edu}_i] + b_{\text{age}\times\text{edu}}[\text{age}_i \, \text{edu}_i] + b_{\text{state}}[\text{state}_i] + b_{\text{region}}[\text{region}_i]$$
$$y_i \sim \text{bernoulli}\left(p_i\right).$$

The dataset was obtained from PosteriorDB (Magnusson et al., 2022).

**Bradley-Terry (BT-Tennis)**    BT-Tennis is a Bradley-Terry model for estimating the skill of professional tennis players used by Giordano *et al.* (2023). The model is described as

$$\sigma \sim \mathcal{N}_+ (0, 1)$$
$$\theta \sim \mathcal{N}\left(\mathbf{0}, \sigma^2 \mathbf{I}\right)$$
$$p_i \sim \theta[\text{win}_i] - \theta[\text{los}_i]$$
$$y_i \sim \text{bernoulli}\,(p)\,,$$

where $\text{win}_i$, $\text{los}_i$ are the indices of the winning and losing players for the $i$th game, respectively. While we subsample over the games $i = 1, \ldots, N$, each player's involvement is sparse in that each player plays only a handful of games. Consequently, the subsampling noise is substantial. Therefore, we use a larger batch size of 500. Similarly to Giordano *et al.* (2023), we use the ATP World Tour data publically available online [1].

**Autoregression (AR-ecg)**    AR-ecg is a linear autoregressive model. Here, we use a Student-t likelihood as originally proposed by Christmas & Everson (2011). While they originally imposed an automatic relevance detection prior on the autoregressive coefficients, we instead set a horseshoe shrinkage prior (Carvalho *et al.*, 2009, 2010). Since the horseshoe is known to result in complex posterior geometry, this should make the problem more challenging. The model is described as

$$\alpha_d = 10^{-2}, \quad \beta_d = 10^{-2}, \quad \alpha_d = 10^{-2}, \quad \beta_d = 10^{-2},$$
$$d \sim \text{gamma}\,(\alpha_d, \beta_d)\,,$$
$$\sigma^{-1} \sim \text{inverse-gamma}\,(\alpha_\sigma, \beta_\sigma)\,,$$
$$\tau \sim \text{cauchy}_+ (0, 1)\,,$$
$$\lambda \sim \text{cauchy}_+ (\mathbf{0}, \mathbf{1})\,,$$
$$\theta \sim \mathcal{N}\,(0, \tau\,\text{diag}\,(\lambda))$$
$$y[n] \sim \text{stduent-t}\,(d, \theta_1 y[n-1] + \theta_2 y[n-2] + \cdots + \theta_P y[n-P], \sigma)\,,$$

where $d$ is the degrees-of-freedom for the Student-t likelihood, $\text{cauchy}_+$ is a half-Cauchy prior.

For the dataset, we use the long-term electrocardiogram measurements of Jager *et al.* (2003) obtained from Physionet (Goldberger *et al.*, 2000). The data instance we used has a duration of 23 hours sampled at 250 Hz with 12-bit resolution over a range of $\pm 10$ millivolts. During the experiments, we observed that the hyperparameters suggested by Christmas & Everson are sensitive to the signal amplitude. Therefore, we scaled the signal amplitude to be $\pm 10$.

---

[1] https://datahub.io/sports-data/atp-world-tour-tennis-data

# H    Additional Experimental Results

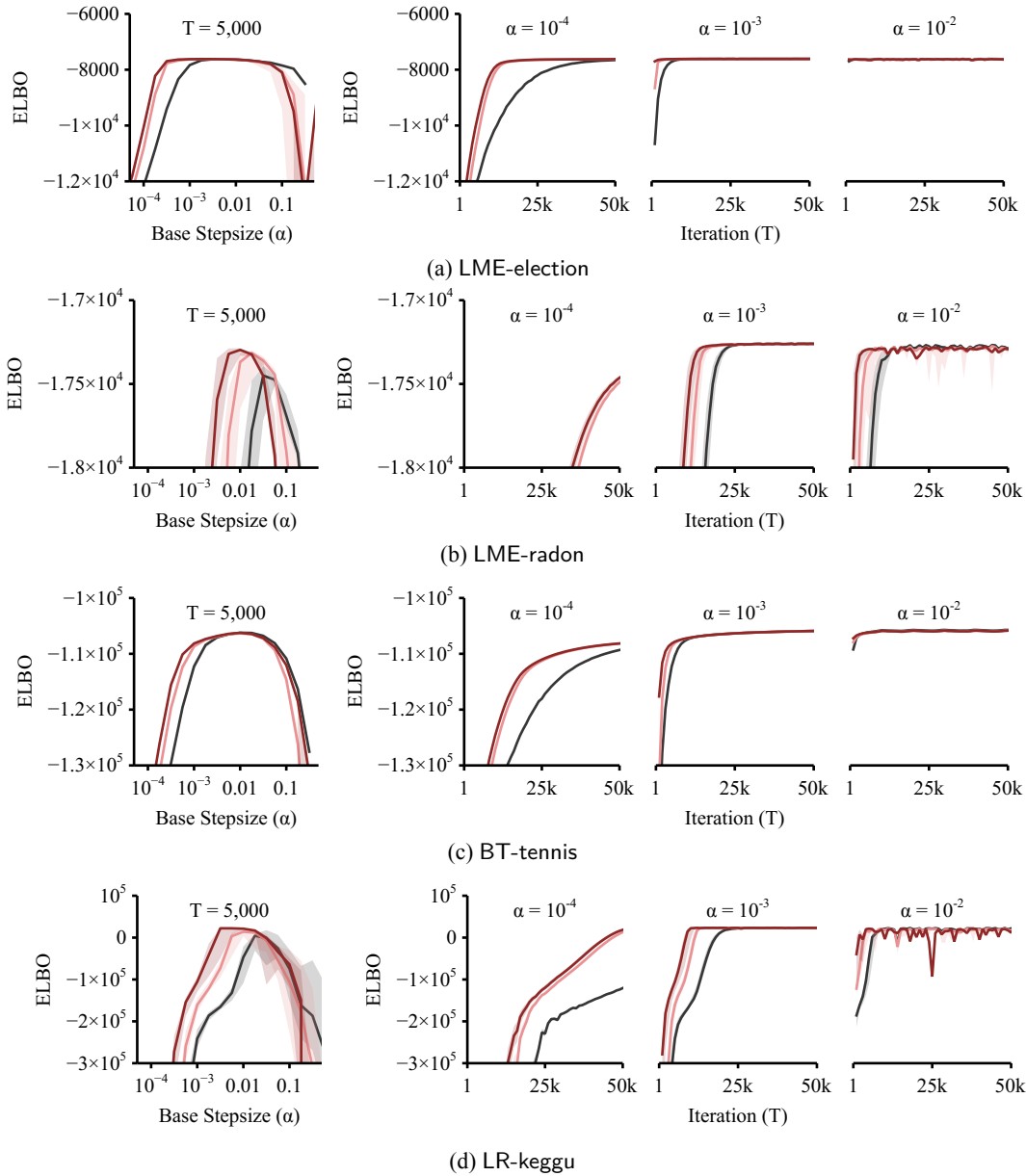

(a) LME-election

(b) LME-radon

(c) BT-tennis

(d) LR-keggu

Figure 5: **BBVI convergence speed (ELBO v.s. Iteration) and robustness against stepsize (ELBO at $T = 50,000$ v.s. Base stepsize).** The error bands are the 80% quantiles estimated from 20 (10 for AR-eeg) independent replications. The initial point was $m_0 = 0, C_0 = I$.

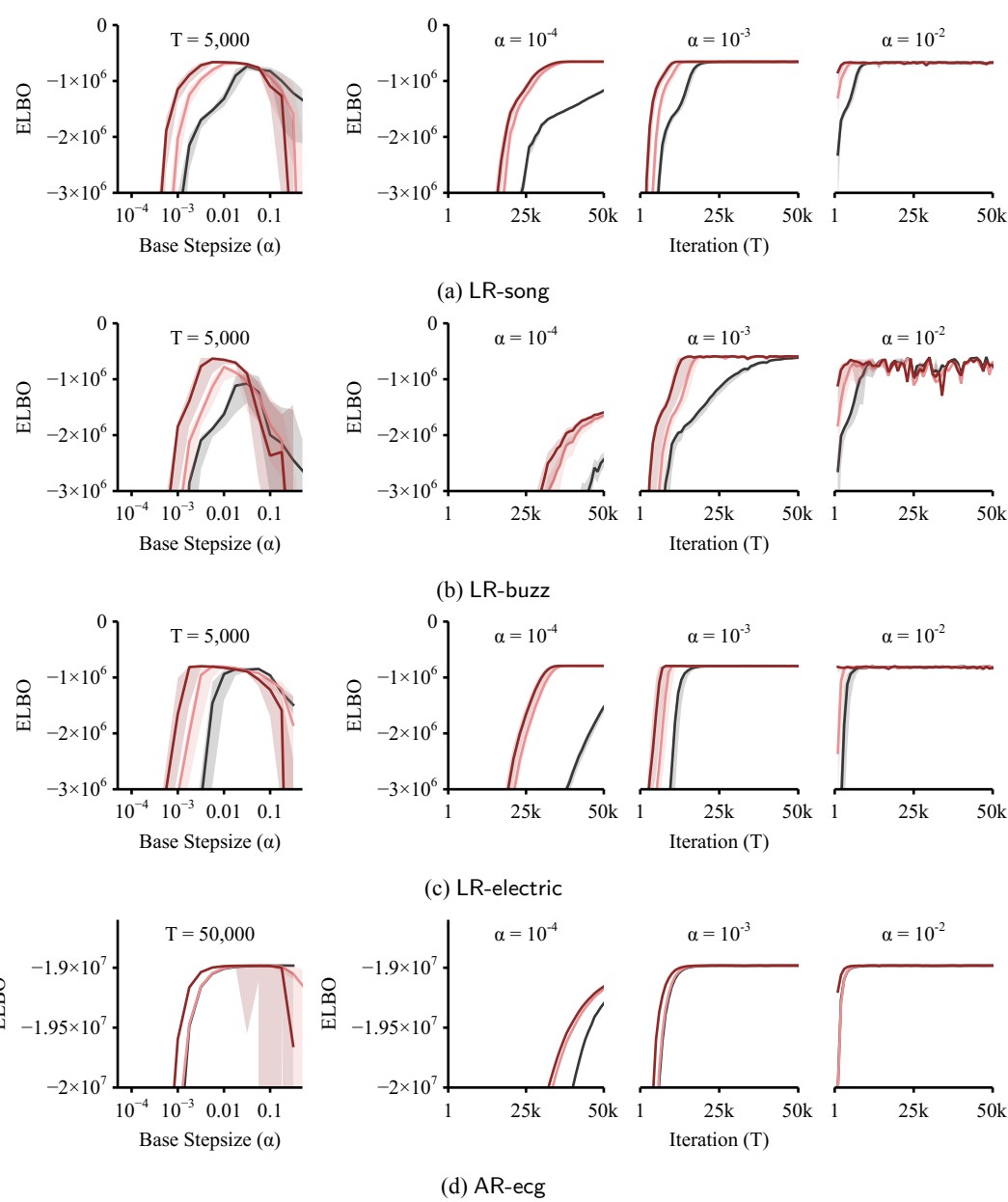

Figure 6: **BBVI convergence speed (ELBO v.s. Iteration) and robustness against stepsize (ELBO at $T = 50,000$ v.s. Base stepsize).** The error bands are the 80% quantiles estimated from 20 (10 for AR-eeg) independent replications. The initial point was $\boldsymbol{m}_0 = \boldsymbol{0}, \boldsymbol{C}_0 = \boldsymbol{I}$.

