# OpenReview forum: "On the Convergence of Black-Box Variational Inference"
_NeurIPS.cc/2023/Conference — NeurIPS 2023 poster_

### Official Review · Reviewer_TwpX · 2023-07-06

**Soundness:** 4 excellent
**Presentation:** 3 good
**Contribution:** 3 good
**Rating:** 7
**Confidence:** 3

**Summary:**

The paper proves convergence results for black-box variational inference (BBVI) with ordinary stochastic gradient descent (SGD) and proximal SGD under several assumptions: it considers only the reparameterization gradient setting with the location-scale variational family (in particular, mean-field and Cholesky parameterization) and a symmetric base distribution (with other mild requirements); it also requires an assumption involving the reparameterized gradient and the diagonal conditioner of the scale matrix and another one about the growth of the bijector. As an improvement over many previous works, it does not need to assume bounded domain, bounded support, or regularity conditions about the evidence lower bound (ELBO) directly.
The paper notes that the theoretical results show that for the location-scale family, nonlinear scale parameterizations are suboptimal (but widely used in practice).
This claim is tested experimentally on a synthetic and realistic problems. The empirical results confirm this claim and suggest that proximal methods converge faster.

**Strengths:**

The paper improves on previous convergence results on BBVI by lifting assumptions needed in previous work. It even goes further by empirically testing the theoretical prediction of linear scale parameterization being superior. Therefore it is interesting and relevant to the variational inference community.

The presentation of the paper is mostly clear and clean despite the technical content. In particular, highlighting the assumptions in the text is very helpful for the reader.

There is a lot of information about the experimental setup in the appendix and the code is available, which is good for reproducibility.

**Weaknesses:**

The main weakness of the paper is how it deals with the limitations: while it is good that the assumptions are highlighted in the text, there are more limitations that should be pointed out. For example, the paper only deals with reparameterization gradient BBVI, which is not mentioned in the abstract. In a similar vein, the title oversells the paper by a lot, making it seem as though it solved a much more general problem. (If the next paper generalizes these results or lifts some of the assumptions, should they give it the same title?) The paper also lacks a limitations section.

Another weakness is that the experimental evaluation (for the most part) uses the Adam optimizer (and a proximal version) even though the theoretical results concern standard and proximal SGD. Furthermore, it explores the combinations Adam&linear, Adam&nonlinear, ProxAdam&linear, but not ProxAdam&nonlinear. I think it would be good to report the results of this combination too, in order to get a more complete picture.

Finally, the results are quite technical, with a lot of variables involved in the bounds and there seem to be some errors in Theorem 3 (see my question to the authors below). It would be good the collect all the variable names and their meaning (or the position of their first occurrence in the text) in one table, so the reader can look up the meaning in one place. As a minor point, it would also help if expected values could include brackets around their arguments, to clarify their scope.

**Questions:**

- What is meant by the "infinite sum structure" in line 115?
- Theorem 3 claims that the iterates generated by BBVI include an $\epsilon$-stationary point. I would have expected a statement of the form "the iterates will be $\epsilon$-stationary points from some point onwards". Does that mean that BBVI will usually move away from the stationary point again?
- In Theorem 3, $\kappa$ is defined but not used, $\gamma$ and $M$ are used but not defined (at least I failed to find their definition) and $C$ takes sometimes one and sometimes two arguments. Could you clarify the statement of the theorem?
- Could you define the "proximal operator" (line 209)? It is defined for the specific setting, but not what it means in general. Some intuition would also be helpful.

Typos:
- line 3: comma should go after the parentheses
- line 14: "box" missing
- line 60: did you mean "sans-serif"?
- line 212: remove "this"
- line 230: Theorem 12 is not in the main part of the paper.

**Limitations:**

Limitations (in the form of assumptions) are mentioned in the paper, but the list is incomplete (see "Weaknesses"). It would be good if all limitations were collected in one place in the paper.

---

> ### Author Rebuttal · Authors · 2023-08-08
>
> Thank you for review.
>
> > The main weakness of the paper is how it deals with the limitations: while it is good that the assumptions are highlighted in the text, there are more limitations that should be pointed out. For example, the paper only deals with reparameterization gradient BBVI, which is not mentioned in the abstract.  In a similar vein, the title oversells the paper by a lot, making it seem as though it solved a much more general problem. (If the next paper generalizes these results or lifts some of the assumptions, should they give it the same title?)
>
> Thank you for the suggestion. We will add this point in the abstract and introduction. We have used of BBVI to mean “BBVI with reparameterization gradients” primarily because we are following recent work on the topic e.g., Domke (2019,2020), Kim et al (2023), Hoffman and Ma (2020), Regier et al. (2017), none of which use score gradients. Nevertheless, we agree that reducing the scope of the title would be more appropriate. We thus propose to change the title to: “On the Convergence and Scale Parameterization of Black-Box Variational Inference.” Please let us know if the reviewer has better suggestions.
>
> > The paper also lacks a limitations section.
>
> We originally had a separate limitation section, but had to remove it due to space issues. We will add it back in the next version.
>
> > Another weakness is that the experimental evaluation (for the most part) uses the Adam optimizer (and a proximal version) even though the theoretical results concern standard and proximal SGD. Furthermore, it explores the combinations Adam&linear, Adam&nonlinear, ProxAdam&linear, but not ProxAdam&nonlinear. I think it would be good to report the results of this combination too, in order to get a more complete picture.
>
> We didn’t include ProxAdam&nonlinear because proximal SGD removes the need to use nonlinear conditioners. We could have included it, but didn’t see the point.
>
> > It would be good the collect all the variable names and their meaning (or the position of their first occurrence in the text) in one table, so the reader can look up the meaning in one place. As a minor point, it would also help if expected values could include brackets around their arguments, to clarify their scope.
>
> Thanks for the suggestion. We will add a separate nomenclature section in the future version.
>
> > What is meant by the "infinite sum structure" in line 115?
>
> We agree that we should have explained this. In the SGD literature, “finite sum” describes an objective function that can be represented as an infinite sum of functions, e.g. $\frac{1}{N} \sum_{n=1}^N f_n(x)$. In contrast, the VI objective cannot be represented as such an objective, as it is an expectation over an infinite number of points. To contrast with the finite sum setting, this setup has been called the “infinite sum.” We will add an explanation and citations to this in the next version.
>
> > Theorem 3 claims that the iterates generated by BBVI include an
> -stationary point. I would have expected a statement of the form "the iterates will be
> -stationary points from some point onwards". Does that mean that BBVI will usually move away from the stationary point again?
>
> This is a limitation of our current understanding of SGD in the nonconvex smooth setting in general, not a problem specific to BBVI. Most of the known analyses prove that the average gradient norm over the iterates is bounded. This in turn implies that $ \min_t || \nabla F (x_t) || < \mathbb{E}_t || \nabla F (x_t) || $. Thus one can only say that the trajectory includes a stationary point (the point achieving the minimum norm.)
>
> > In Theorem 3, gamma is defined but not used, kappa and M are used but not defined (at least I failed to find their definition) and C takes sometimes one and sometimes two arguments. Could you clarify the statement of the theorem?
>
> Thanks for pointing out that $\kappa$ is not used. Here $gamma$ is the step size, $M$ is the number of samples, and $C$ is used as a general placeholder for arbitrary constants. As suggested before, we will add a separate nomenclature table to resolve the confusion.
>
> > Could you define the "proximal operator" (line 209)? It is defined for the specific setting, but not what it means in general. Some intuition would also be helpful.
>
> Thank you for the suggestion. Yes, we should have included an explanation.
>
> In general, a proximal operator (or proximity operator) is a tool commonly used in optimization. For a convex function $h$ it is generally described as $\mathrm{prox}_{\gamma, h}(x) = \arg\min_z \frac12 \| x - z \|^2 + \gamma h(z)$, where $\gamma$ is the stepsize. We want to find a point close to $x$ and simultaneously minimize the function $h$. While it might not be obvious, this operator is a generalization of the gradient descent update $x_t - \gamma g_t$, which can be obtained by setting $h$ to be the first-order linearization of the objective function.
>
> A typical use-case of proximal operators is when $h$ is non-smooth or even non-differentiable. In those cases, the proximal operator may have a closed-form expression and favorable convergence properties than, say, subgradient descent. In our case, the proximal operator is used to circumvent the fact that the entropy term is non-smooth.
>
> We will add this explanation in the next version.

---

> > ### Comment · Reviewer_TwpX · 2023-08-15
> >
> > Thank you for the response. It clears up a lot of things. I'm glad that you've decided to pick a more specific title and to re-add a limitations section.
> >
> > I noticed that you did not address the point regarding Adam in the experiments vs standard SGD in the experiments. Would you like to comment on that?

---

> > > ### Author Response · Authors · 2023-08-16
> > > **Response**
> > >
> > > Thank you very much for the engagement. We are glad that our response has cleared up the discussion.
> > >
> > > Sorry that we missed answering the point on Theory with SGD v.s. Experiments with Adam. In our original submission, we conduct two types of experiments:
> > > - (a) A "controlled" synthetic experiment where the assumptions of the paper can be met perfectly (Section 4.1, Figure 3), and
> > > - (b). "realistic" experiments where the settings are closer to how BBVI is done in practice (Section 4.2, Figure 4).
> > >
> > > In (a), we used SGD, while in (b), we used Adam. Thus, we provide experimental results in both theoretical and practical extremes. We observe that, in (b), due to the fact that Adam handles non-smoothness quite well (although this is not well understood), the differences between parameterizations become narrower than in (a). But we still observe an effect that is in line with our theory and the controlled experiments in (a).
> > >
> > > We hope this answers the original comment and we more clearly state our experimental intent in the future version. Please let us know anytime if there are further questions about our work.

---

> > > > ### Comment · Reviewer_TwpX · 2023-08-16
> > > >
> > > > Thank you for that clarification. This makes sense to me now. As a result of the rebuttal, I have raised my rating from 6 to 7.

---

### Official Review · Reviewer_MsbX · 2023-07-06

**Soundness:** 4 excellent
**Presentation:** 3 good
**Contribution:** 3 good
**Rating:** 7
**Confidence:** 3

**Summary:**

The authors analyze the smoothness and convexity of the ELBO under different parameterizations (linear vs nonlinear) of the scale for location-scale families, building on the work of Domke (2020). This enables convergence analysis for BBVI with standard and proximal stochastic gradient descent. Their main findings are that 1) the energy term of the ELBO is smooth under certain conditions (Theorem 1, ii) on the diagonal conditioner $\phi$ and reparameterization gradient, and 2) nonlinear diagonal conditioners break (strong) convexity of the energy, affecting SGD convergence rates, and 3) establishing rates for obtaining $\epsilon$-optimal solutions in BBVI for both convex and nonconvex energy terms.


**Strengths:**

* The work extends previous results meaningfully: Domke (2020) considered the case of a linear diagonal conditioner, while this work focuses on nonlinear ones while including the linear case for completeness.
* Beyond showing convergence merely occurs (per the title), rates are established both for standard SGD iterates (Theorem 3) and proximal SGD in the case of a convex $f$.
* The variational family considered (location-scale) encompasses a wide range of distributions usually used for VI; one additional step of  generalizability is the use of the bijector $\psi$ to make the work relevant to ADVI in general.
* The work has theoretical and practical merit, advising users on the tradeoffs between linear and nonlinear diagonal conditioners along with providing clearly established lemmas, assumptions and proofs that further work might build upon.


**Weaknesses:**

* The story is a bit disjointed. Assumption 4 and Theorem 3 might be better placed directly after Example 4 to complete a story about convergence of BBVI for nonlinear $\phi$ and general nonconvex $f$ that satisfies the smoothness conditions (among others). This might have even been a logical stopping point for a paper with the given title. Theorem 2 seems better placed nearby the section on Proximal SGD, as it is precisely the convexity of $f$ that Section 3.4 requires.
* The exponential diagonal conditioner (line 108-109), commonly used in practice,  does not satisfy the 1-Lipschitz assumption (Theorem 1, (ii)). I put this as a weakness, but maybe this point should be emphasized instead and used to evangelize the merits of softplus, which evidently does satisfy this assumption.
* The case where $f$, which is usually $-\log p(x,z)$ in this work when $\psi$ is the identity, is convex in $z$ is uncommon in many practical problems of significance for VI when the density $p(x,z)$ is multimodal. This diminishes the significance of Theorem 2 (breaking convexity) and Section 3.4.


**Questions:**

* The summary of contributions (lines 51-58) seems overly informal to me. What is a “full” guarantee? What converges? “Precisely as used in practice” may mean different things to different people. “Suboptimal” in what sense (item #2)? Maybe more precision would be better here.
* The formatting of lines 67-70 is odd, and perhaps these quantities should just be defined in-line.
* In line 102, I believe an additional “diag” is missing: should it be $diag(\phi(s)) = diag(\phi(s_1),\dots, \phi(s_d))$?
* Line 115, “inifinite” is a typo. This taxonomy discussion I find detrimental to the paper, as it creates more confusion than clarity; Figure 1 and its caption could be removed altogether.
* Is the distinction between the LHS and RHS of the equation after line 127 that of a “total derivative” vs a partial derivative with respect to $\lambda$?
* Line 130-132 has a typo in phrasing “have been”, “alienating”
* The proof of Theorem 1 seems to imply that $f$, in addition to being smooth, is twice-differentiable to compute the Hessian. If this isn’t implied by some other conditions, it should be stated explicitly perhaps for completeness.
* The statement of Theorem 2 differs in the main paper and the supplement, this should be corrected.
* Line 174, “becomes” is a typo


**Limitations:**

The theoretical contributions make clear the assumptions that must be met for the results to hold. As stated, the 1-Lipschitz condition for $\phi$ could be discussed more.

---

> ### Author Rebuttal · Authors · 2023-08-08
>
> Thank you for your review.
>
> > The story is a bit disjointed. Assumption 4 and Theorem 3 might be better placed directly after Example 4 to complete a story about convergence of BBVI for nonlinear
>  and general nonconvex $f$ that satisfies the smoothness conditions (among others). This might have even been a logical stopping point for a paper with the given title. Theorem 2 seems better placed nearby the section on Proximal SGD, as it is precisely the convexity of
>  that Section 3.4 requires.
>
> Thank you for the suggestions. We will reorganize our paper in the next version accordingly.
>
> > The exponential diagonal conditioner (line 108-109), commonly used in practice, does not satisfy the 1-Lipschitz assumption (Theorem 1, (ii)). I put this as a weakness, but maybe this point should be emphasized instead and used to evangelize the merits of softplus, which evidently does satisfy this assumption.
>
> We point out that Theorem 2 does say something about non-Lipschitz conditioners; the exp conditioner will also result in a slower convergence rate since it breaks strong convexity. Given the breadth of this result, we thought it didn’t matter much which conditioner is used; any non-linear conditioner is bad. Nevertheless, we will add the suggested comment in the limitations.
>
> > The case where $f$, which is usually $- \log p(x, z)$ in this work when $\psi$ is the identity, is convex in $z$ is uncommon in many practical problems of significance for VI when the density $p(x, z)$ is multimodal. This diminishes the significance of Theorem 2 (breaking convexity) and Section 3.4.
>
> We believe that the significance of our theory should be taken qualitatively rather than quantitatively. Our results say that when the posterior is “easy” such that the landscape is Gaussian, nonlinear conditioners will fail to take advantage of this, unlike linear conditioners. If we extrapolate this intuition to posteriors that are non-Gaussian but not too much, it is still conceivable that linear conditioners will result in faster convergence. It is also known that, for large datasets, many Bayesian posteriors become Gaussian as a consequence of the Bernstein von Mises theorem. Thus we believe our results do explain the performance difference of our experiments, where none of the problems are provably Gaussian (strongly log-concave.)
>
> > Is the distinction between the LHS and RHS of the equation after line 127 that of a “total derivative” vs a partial derivative with respect to $\lambda$
>
> Sorry for the confusion. We should have added a written description. The RHS is derivative with respect to $\lambda$, while the LHS is the composition of $t_{\lambda}(u)$ and the gradient of $f$ denoted as $\nabla f$. That is, $\nabla f \circ t_{\lambda}(u)$.
>
> > The summary of contributions (lines 51-58) seems overly informal to me. What is a “full” guarantee? What converges? “Precisely as used in practice” may mean different things to different people. “Suboptimal” in what sense (item #2)? Maybe more precision would be better here.
>
> Thank you for the comments. We will be more precise in the contribution summary and state that we do not assume any unrealistic modifications to the algorithm, such as bounded domain or bounded gradient. For the “suboptimal,” we will state that the nonlinear conditioners break strong convexity.
>
> > The proof of Theorem 1 seems to imply that $f$, in addition to being smooth, is twice-differentiable to compute the Hessian. If this isn’t implied by some other conditions, it should be stated explicitly perhaps for completeness.
>
> Thank you very much for catching this! Indeed we missed that assumption. We will add it in the next version.
>
> > The statement of Theorem 2 differs in the main paper and the supplement, this should be corrected.
>
> Sorry for the confusion. We will fix this in the next version.

---

> > ### Comment · Reviewer_MsbX · 2023-08-16
> > **Rebuttal response**
> >
> > I thank the authors for the detailed response. Most of my points were minor and I am glad to see some will be addressed by the authors for clarity. The title change and the limitations section will help practitioners find this work and understand how the results affect implementations of BBVI.

---

### Official Review · Reviewer_C3v1 · 2023-07-07

**Soundness:** 3 good
**Presentation:** 3 good
**Contribution:** 2 fair
**Rating:** 7
**Confidence:** 3

**Summary:**

This paper tries to prove convergence of BBVI algorithm in a more general setting compared to previous work: the main being that the target needing to be only log-smooth and not needing to be log-concave, which means that the objective can be non-convex. The paper claims that non-linear parameterizations of the variational parameters (the mean, the cholesky factor L for full rank matrix and scale for mean field), often used in practice can break strong convexity, even if the target is log-concave and thus plain SGD with non-linear conditioning is sub-optimal while proximal SGD with linear parameterization can give the fastest convergence for BBVI.

**Strengths:**

1. The paper is solid and covers a lot of theory behind VI, Figure 1 is quite informative.
2. The figures are great and descriptive and support the theory and text.
3. I am not good with theory, but I found the theorems and inequalities fine
4. Covers relevant and contemporary literature quite well, although a paper with similar ideas and claims recently appeared on arxiv.


**Weaknesses:**

1. I think, the paper is trying to unify many different parameterizations, and so at times the paper has become a bit harder to read, and in some places, authors can motivate more why and how certain theorems can have pratical impact, where things are generally followed and discussion is only theoretical and where theorems have a practical impact.
2. Then my main concern with this work, is that its claims can be easily misinterpreted, when readers see: 'BBVI converges', especially when there is no general consensus of what black box means, I think BBVI was first used as an acronym by the Ranganath 2013 paper which actually used score gradients. This nomenclature of algorithms is unfortunate because many new readers also think of Neural Networks powered inference algorithms, when they hear Black-box VI and assume Normalizing flows and Neural Network based Variational inference also as BBVI. With the recent explosion of methods, where a user can practically choose any divergence measure(any member of the f-divergence family) and any approximating family and use MC samples and auto diff to estimate gradients for inference, while this paper is narrow in the scope that it only considers exclusive KL, well behaved log-concave Gaussian (loc-scale) variational family and RP gradients, it will be unfair to say that 'BBVI converges', and therefore I strongly suggest authors to change the title of the paper.
3. In results section, it seems a bit odd and unsettling to me that the linear parameterization is claimed to be better than non-linear in all cases. Another submission in this conference, claims that non-linear(log) parameterization is theoretically more stable (also in Domke 2019), and shows some practical experiments confirming the same.

**Questions:**

1. If I understood correctly, the object of treatment in this paper is the objective function and not the gradient, both Theorems 1 and 2 are for the objective rather, I wanted to know why did the authors concentrate on objective and not the gradient itself.
2. Is it expected that the gradients will follow the same behaviour as objective which is the main object of treatment in this work, especially when there is a non-linear transformation involved in the parameterization, i.e. $ \nabla_{\lambda}f(\mathbf{t}_{\lambda}(\mathbf{u})) $ is different from $\nabla f$ ?
3. Does the work cover the case when gradients are computed with mini-batching which brings another source of stochasticity in addition to randomness of base distribution ?
4. Maybe the authors can also address, how their work is different/similar to the one in 'Provable convergence guarantees for black-box
variational inference' on arxiv, how their claims match to it ?
5. Why was the exp conditioner/transformation not used in expts. for Figure 4, when it is the most popular one in my opinion ?
6. I did not like this statement so much: 'This would put mean-field BBVI in a regime comparable to the best-known convergence rates in
approximate MCMC' since MCMC is aymptotically exact while MFVI is not and can be arbitrarily different from target.

**Limitations:**


Minor
1. Typo in line 115:'inifinite'
2. For the uninitiated, maybe introduce $L_{h}$ as Lipschitz smooth, if it does .. ?
3. Maybe explain what do all the constants $L_{f}, L_{h}, L_{s}$ mean at the beginning of Sec 3. I guess $L_{f}, L_{h}$ mean smothness constants for likelihood/energy part and entropy part in the objective respectively,

I am overall inclined in favour of accepting the paper given some of the questions and concerns are addressed by the authors. I need also a bit more time in analysing this work and recent similar work, where some of the claims are similar but some are different.

---

> ### Author Rebuttal · Authors · 2023-08-08
>
> Thank you for your review.
>
> > I think, the paper is trying to unify many different parameterizations, and so at times the paper has become a bit harder to read
>
> We agree that the paper is quite dry in the submitted form. Due to the page limit, we had to remove many of our results' commentaries. We will add more practical discussions of our results in the next version.
>
> > Then my main concern with this work, is that its claims can be easily misinterpreted, when readers see: 'BBVI converges', especially when there is no general consensus of what black box means, I think BBVI ... actually used score gradients.
>
> In our defense, much recent work on BBVI with Monte Carlo gradients does seem to use “BBVI” synonymously. Recent papers such as the work of Domke (2019,2020), Hoffman and Ma (2020), Regier et al. (2017), Manushi et al. (2022) all use the term BBVI despite none of these works considering the score gradient. We are happy to use more general terms such as Monte Carlo VI and clarify this in the abstract. However, we’d also like to find some middle ground here and not deviate substantially from conventional technical vocabulary norms set in the recent literature,
>
> >  I strongly suggest authors to change the title of the paper.
>
> We agree that the title might be perceived overly broadly. We propose to change the title to: “On the Convergence and Scale Parameterization of Black-Box Variational Inference.” Please let us know if the reviewer has better suggestions.
>
> > In results section, it seems a bit odd and unsettling to me that the linear parameterization is claimed to be better than non-linear in all cases. Another submission in this conference, claims that non-linear(log) parameterization is theoretically more stable (also in Domke 2019)
>
> Some of the authors have also seen that paper during bidding, although we couldn’t read it (as we didn’t bid for it due to the apparent conflict of interest). Thus, unfortunately, it would be impossible and inappropriate for us to comment on their results. However, Thm 2 is quite clear; you lose strong convexity with non-linear parameterizations, and thus one loses the guarantee to converge fast. To us, it was thus unsurprising that the linear param. did well in the experiments. Furthermore, we are unaware that Domke (2019) discusses non-linear params. His recent paper ($\S$ 5; Domke et al., 2023) agrees with our theoretical analysis: “*An alternate parameterization would also have implications for the (strong) convexity of the ELBO.*”
>
> In practice, the performance strongly depends on the choice of initialization. So this might explain the different conclusions. For instance, when initialized in the non-smooth regime (small initial scale), the linear parameterization starts by diverging (visualized in Domke 2020 Fig 1), resulting in slow convergence. However, as shown in Fig 3 in our paper and Domke (2020), proximal SGD fixes this sensitivity.
>
> > If I understood correctly, the object of treatment in this paper is the objective function and not the gradient ... I wanted to know why did the authors concentrate on objective and not the gradient itself.
>
> For studying the properties of SGD, one needs to analyze both the objective and gradient estimator. In particular, the landscape of the objective (smoothness, convexity) often determines the convergence rate. We thus focus on the properties of the ELBO.
>
> > Is it expected that the gradients will follow the same behaviour as objective which is the main object of treatment in this work?
>
> We kindly request clarification for this question.
>
> > Does the work cover the case when gradients are computed with mini-batching ... ?
>
> Yes, this should be fairly trivial to do whenever the linearity of the variance is used. We will add a discussion on how to do this in the future version. We also found that more insight could be drawn in the doubly stochastic setting, which we are attempting to study in a follow-up paper.
>
> > Maybe the authors can also address, how their work is different/similar to the one in 'Provable convergence guarantees for black-box variational inference' on arxiv, how their claims match to it ?
>
> We indeed saw that paper. Before we comment on the paper at all, note that the paper is concurrent work (potentially a concurrent NeurIPS submission), which we need to tread carefully about considering this paper here, as the NeurIPS FAQ suggests we should generally not do (see the “policy on comparisons to recent work”).
>
> While we emphasize the policy above, we are willing to contrast the two. Our paper was more interested in the effect of parameterizations, while that paper was more focused on proving the convergence of SGD with quadratic variance estimators, which were studied in Domke (2019). The only overlap is for the convergence of proximal SGD with a decreasing stepsize for strongly convex ELBO, for which the complexity matches perfectly: $\mathcal{O}(\kappa^2 \frac{1}{\epsilon})$.
>
> > Why was the exp conditioner/transformation not used ... ?
>
> To our knowledge, the softplus conditioner is also popularly used (see Tab 1 of Kim et al. 2023.) Furthermore, from the conclusions of Thm 2, we expected the performance of the softplus to be representative of all nonlinear conditioners.
>
> > I did not like this statement so much: 'This would put mean-field BBVI in a regime comparable to ... approximate MCMC' since MCMC is aymptotically exact while MFVI is not and can be arbitrarily different from target.
>
> Here, we’d like to emphasize the use of the word approximate, and will make this clearer in the text. The fastest convergence rates in MCMC are achieved with “approximate” MCMC methods, which are asymptotically biased, such as unadjusted Langevin. The cited paper indeed discusses the unadjusted Langevin algorithm, which is an approximate MCMC method.
>
> > For the uninitiated, maybe introduce Lipschitz smooth ...
>
> Thanks for the suggestion. We will add more discussion on these assumptions in the future version.

---

> > ### Comment · Reviewer_C3v1 · 2023-08-16
> > **Reply to rebuttal**
> >
> > Thanks to the authors for point by point response to my questions. I only wanted to emphasize that treatment of both objective and gradients is important in analysis for convergence, which they agree with. I like the new proposed title much more than the previous one which was my main concern and I must point out that other reviewers also felt the same. Then, I also felt that the authors tried too hard in overselling itself a bit too much in places when they actually didn't need to, which other reviewers also noticed. I am other wise happy to revise my rating and recommend acceptance. I think I am also satsfied with their response in comparison to Domke's old and recent work. Maybe they could have a joint session with the authors if the both papers are accepted. I remember a similar thing happening in last year NeurIPS when two accepted papers had conflicting results and conclusions, which is not necessarily the case here. From practise, I know that BBVI is so much dependent on initialization that it can be hard to say a general statement about parameterisations, and this is talking about exclusive KL, the other commonly used divergence objectives are even more wild.

---

### Official Review · Reviewer_gnfU · 2023-07-08

**Soundness:** 4 excellent
**Presentation:** 4 excellent
**Contribution:** 2 fair
**Rating:** 7
**Confidence:** 4

**Summary:**

The paper establishes the first complete convergence result of BBVI as it is used in practice. It is shown that a linear parametrization of the covariance leads to better convergence rates, which is confirmed in some numerical experiments. Finally, a proximal version for BBVI due to Domke et al 2020 is analyzed and shown to perform favorably in experiments to Adam.

**Strengths:**

- The paper is well-written and clearly a lot of time was spent by the authors to make the results look clean and polished.
- The proposed proximal scheme for BBVI is implemented in the probabilistic programming language Turing, which may be of interest to the community once the code is released.


**Weaknesses:**

1) The results and experiments are incremental (but nontrivial) extensions of what is presented in (Domke 2020).

2) The paper missed to cite the recent line of works
* https://arxiv.org/abs/2205.15902
* https://proceedings.mlr.press/v202/diao23a.html
* https://proceedings.mlr.press/v202/diao23a.html
which give very strong convergence results for a BBVI-like algorithm by interpreting it as a Wasserstein gradient flow.  A theoretical (and maybe even practical) comparison to these recent works would make the paper stronger.

3) It is known that BBVI is inferior to natural-gradient algorithms which exploit the KL / Fisher-geometry (https://arxiv.org/abs/2107.04562).  This line of works could be mentioned in the introduction.


**Questions:**

1) When using the linear parametrization, could there be an issue that the scale parameter becomes negative? Or is this somehow handled by the proximal operator of the entropy?

2) Do the proposed convergence results also carry over to ProxGen? Perhaps it is out of scope for this paper, but maybe a comment in the experiments section could be helpful.

3) How well does the method work for modern neural networks  (ResNets, transformers)? There is many claims in the community that variational inference doesn't work well in these settings (e.g. worse performance than MAP inference). It would be highly interesting to see whether the proximal scheme solves these problems.

**Limitations:**

All limitations are addressed.

---

> ### Author Rebuttal · Authors · 2023-08-08
>
> Thank you for your review.
>
> > The paper missed to cite the recent line of works
>
> Thank you for the suggestions. We will add them to the next version.
>
> > It is known that BBVI is inferior to natural-gradient algorithms which exploit the KL / Fisher-geometry (https://arxiv.org/abs/2107.04562).  This line of works could be mentioned in the introduction.
>
> Given the importance of NGVI, we agree to include it in the introduction and discussion sections. However, we would note that BBVI is a broader algorithm in the sense that one can use variational distributions outside of the exponential family, which our theory does include, and amortized variational families, which we plan on working on next.
>
> > When using the linear parametrization, could there be an issue that the scale parameter becomes negative? Or is this somehow handled by the proximal operator of the entropy?
>
> Yes, the proximal operator ensures that the scale is never negative. Furthermore, using projection operators as suggested by Domke (2020) is also an effective way of ensuring this. In practice, however, choosing a stable initial point (large scale) with a small enough step size appears to be enough to ensure convergence, though there is still room for investigation on whether it is a robust enough choice.
>
> > Do the proposed convergence results also carry over to ProxGen? Perhaps it is out of scope for this paper, but maybe a comment in the experiments section could be helpful.
>
> Unfortunately, ProxGen assumes the problematic “bounded gradient variance” assumption, thus our results cannot be immediately applied. It might be possible to extend their proof to include our setting. As suggested, we will add a discussion about this.
>
> > How well does the method work for modern neural networks (ResNets, transformers)? There is many claims in the community that variational inference doesn't work well in these settings (e.g. worse performance than MAP inference). It would be highly interesting to see whether the proximal scheme solves these problems.
>
> Given the theoretical and empirical results on “pruning” (Trippe et al., 2018; Coker et al., 2022; Huix et al., 2022), we conjecture that the proximal scheme will not fix the problem. Exclusive KL may simply be inappropriate for deep models, although more investigation is certainly needed.
>
> ### References
> Trippe, Brian, & Turner, Richard. 2017. Overpruning in Variational Bayesian Neural Networks. Tech. rept. arXiv:1801.06230. ArXiv.
>
> Coker, Beau, Bruinsma, Wessel P., Burt, David R., Pan, Weiwei, & Doshi-Velez, Finale. 2022. Wide Mean-Field Bayesian Neural Networks Ignore the Data. Pages 5276–5333 of: Proceedings of the International Conference on Artificial Intelligence and Statistics. PMLR.
>
> Huix, Tom, Majewski, Szymon, Durmus, Alain, Moulines, Eric, & Korba, Anna. 2022 (July). Variational Inference of Overparameterized Bayesian Neural Networks: A Theoretical and Empirical Study. Tech. rept. arXiv:2207.03859. ArXiv

---

> > ### Comment · Reviewer_gnfU · 2023-08-17
> >
> > Thanks for the clarifications, I am satisfied with the rebuttal. Just a small remark, natural-gradient methods do not necessarily require the distribution to be an exponential family [1,2].
> >
> > [1] https://arxiv.org/abs/1906.02914
> > [2] https://arxiv.org/abs/2303.04397

---

### Decision · Program_Chairs · 2023-09-21

**Decision:**

Accept (poster)

**Comment:**

All reviewers agreed this paper would make a solid contribution to NeurIPS. In the camera ready, please carefully go through the reviews, your rebuttal, and the discussion, and update the paper accordingly. Some particular notes:

- include noted papers in the literature review
- edit the title to avoid misleading claims
- there was some concurrent work brought up during the review process (I noted that one reviewer brought up a paper with seemingly contradictory claims to the present paper, and another that may have overlap); please include a discussion of these works if reasonable to do so, with a mention that they are concurrent. Note that the neurips policy encourages comparison to concurrent work in the camera ready, and I think it would be beneficial for readers in this instance.
- a comprehensive limitations section
- some suggested reorganizations of the text/results